# Metabolic Heterogeneity, Plasticity, and Adaptation to “Glutamine Addiction” in Cancer Cells: The Role of Glutaminase and the GTωA [Glutamine Transaminase—ω-Amidase (Glutaminase II)] Pathway

**DOI:** 10.3390/biology12081131

**Published:** 2023-08-14

**Authors:** Arthur J. L. Cooper, Thambi Dorai, John T. Pinto, Travis T. Denton

**Affiliations:** 1Department of Biochemistry and Molecular Biology, New York Medical College, 15 Dana Road, Valhalla, NY 10595, USA; t_dorai@hotmail.com (T.D.); pintoj@optonline.net (J.T.P.); 2Department of Urology, New York Medical College, Valhalla, NY 10595, USA; 3Department Pharmaceutical Sciences, College of Pharmacy & Pharmaceutical Sciences, Washington State University Health Sciences Spokane, Spokane, WA 99202, USA; 4Department of Translational Medicine and Physiology, Elson S. Floyd College of Medicine, Washington State University Health Sciences Spokane, Spokane, WA 99164, USA; 5Steve Gleason Institute for Neuroscience, Washington State University Health Sciences Spokane, Spokane, WA 99164, USA

**Keywords:** ω-amidase, l-glutamine, l-glutamate, l-glutamine addiction, glutaminase II, GLS1, GLS2, glutamine transaminase, GTωA pathway, α-ketoglutaramate, α-ketoglutarate, asparaginase II pathway, asparaginase, α-ketosuccinamate

## Abstract

**Simple Summary:**

Many types of cancer cells utilize the common amino acid l-glutamine to maintain their metabolic demands for energy and the nitrogen required for DNA synthesis. Versatility to control energy metabolism from l-glutamine (anaplerosis) is promoted by the use of two distinct pathways. The first (pathway 1; the canonical pathway) is as follows: [l-glutamine → l-glutamate ⇆ α-ketoglutarate → tricarboxylic acid (TCA) cycle]. This pathway contrasts with the much less studied GTωA (glutamine transaminase—ω-amidase or glutaminase II) pathway (pathway 2): [l-glutamine ⇆ α-ketoglutaramate (KGM) → α-ketoglutarate → TCA cycle]. Our prior publications have emphasized the importance of regulation of both pathways, which enables cancer cells to maintain selective metabolic advantages and to conserve their reliance on glucose. This review summarizes the metabolic importance of the GTωA pathway in both cancerous and normal tissues and proposes that anti-cancer strategies, based on inhibition of l-glutamine metabolism, require consideration of both the canonical and GTωA pathways.

**Abstract:**

Many cancers utilize l-glutamine as a major energy source. Often cited in the literature as “l-glutamine addiction”, this well-characterized pathway involves hydrolysis of l-glutamine by a glutaminase to l-glutamate, followed by oxidative deamination, or transamination, to α-ketoglutarate, which enters the tricarboxylic acid cycle. However, mammalian tissues/cancers possess a rarely mentioned, alternative pathway (the glutaminase II pathway): l-glutamine is transaminated to α-ketoglutaramate (KGM), followed by ω-amidase (ωA)-catalyzed hydrolysis of KGM to α-ketoglutarate. The name glutaminase II may be confused with the glutaminase 2 (GLS2) isozyme. Thus, we recently renamed the glutaminase II pathway the “glutamine transaminase—ω-amidase (GTωA)” pathway. Herein, we summarize the metabolic importance of the GTωA pathway, including its role in closing the methionine salvage pathway, and as a source of anaplerotic α-ketoglutarate. An advantage of the GTωA pathway is that there is no net change in redox status, permitting α-ketoglutarate production during hypoxia, diminishing cellular energy demands. We suggest that the ability to coordinate control of both pathways bestows a metabolic advantage to cancer cells. Finally, we discuss possible benefits of GTωA pathway inhibitors, not only as aids to studying the normal biological roles of the pathway but also as possible useful anticancer agents.

## 1. Introduction

It is well known that many cancers utilize glutamine as a source of α-ketoglutarate for energy production, and as a source of nitrogen for DNA synthesis (see Section 9). We discuss here the **canonical** pathway for conversion of l-glutamine to α-ketoglutarate in cancer cells (i.e., l-glutamine → l-glutamate ⇆ α-ketoglutarate). This pathway is initiated by the glutaminase isozyme GLS1 (KGA) and/or its splice variant GAC (and possibly, in some cancer cases, by the glutaminase isozyme GLS2), followed by conversion of l-glutamate to α-ketoglutarate, catalyzed by α-ketoglutarate-dependent transaminases or by redox-dependent glutamate dehydrogenase (GDH). The major intent of the present review is to document the metabolic subtleties of l-glutamine homeostasis in normal and and cancerous cells, with particular reference to an alternative pathway for generating α-ketoglutarate from l-glutamine, namely the **glutamine transaminase-ω-amidase** (**GTωA**) **pathway** (l-glutamine ⇆ α-ketoglutaramate (KGM) → α-ketoglutarate), which, for historical reasons, we previously referred to as the **glutaminase II pathway**. Although we have addressed this topic previously, e.g., [1,2,3,4], we deemed it necessary to re-enforce the biological importance of the GTωA pathway in the context of overall l-glutamine homeostasis. Thus, some of the present communication incorporates previously reviewed material. However, we also include new data and ideas supporting the hypothesis that the GTωA pathway is highly adaptive and integrative with the more widely known canonical pathway of l-glutamine metabolism. To fully appreciate the metabolic significance of the pathways involved in conversion of l-glutamine to α-ketoglutarate, we have provided a historical perspective. In addition, we have highlighted the importance of clarifying enzyme nomenclature related to glutamine transamination. Because most biomedical researchers are not familiar with the GTωA pathway, we then consider the proposed biological roles of this pathway when cells need to adjust energy demands to maintain homeostasis. Finally, we discuss the importance of the GTωA pathway in supplying anaplerotic α-ketoglutarate to cancer cells, providing an advantage to these cells, especially under anaerobic/hypoxic conditions. We include a discussion of the integration of the GTωA pathway with those involving l-asparagine (asparaginase II pathway) and methionine salvage and suggest why inhibitors of the GTωA pathway will have distinct clinical relevance.

## 2. Discovery of Enzymes That Convert l-Glutamine Amide to Ammonium and Notes on Nomenclature

In the 1940s, Jesse Greenstein and colleagues at the National Institutes of Health (NIH) discovered glutaminase activity in liver, kidney, and other organs in rodents. This enzyme, which they named **glutaminase I** [Equation (1)], was shown to be stimulated by phosphate and arsenate [5,6,7]. (***Note the Roman numeral for one***). This enzyme is now usually referred to as glutaminase or phosphate-activated glutaminase (**PAG**). In mammals, including humans, PAG occurs as two mitochondrial isozymes, namely a kidney type (**KGA**, **GLS1**) and its active splice variant (**GAC**) and a liver type (**GLS2**, **GAB**) and its less active, shortened form (**LGA**) [8,9,10,11]. The kidney-type glutaminase, GLS1, is often referred to simply as GLS. The designation GLS1 for the kidney-type glutaminase is used throughout the present review. A K-cDNA probe (i.e., a kidney-type probe) and Western blotting showed that, in humans, the GLS1 gene is located on chromosome 2 and the transcript is present in kidney, brain, heart, placenta, lung, and pancreas [12]. Additionally, an l-cDNA probe (i.e., a liver-type probe) was used to show that in humans, the GLS2 gene is located on chromosome 12 and the transcript is present in liver, brain, and pancreas [12]. The liver enzyme is unusual in that it is feed-forward-activated by its product, ammonium (^+^NH_4_) [13]. In the liver, GLS2 is mainly located in the urea-cycle-containing periportal hepatocytes, allowing incorporation of ammonium, derived from the amide portion of l-glutamine, into urea (e.g., [14]). Glutamine synthetase (glutamate ammonia ligase) resynthesizes l-glutamine in the perivenous hepatocytes from the ammonium that evades periportal urea synthesis [14]. Thus, in view of its metabolic function, recent investigations have suggested that inhibition of hepatic GLS2 may reduce the risk of hyperammonemia and other clinical manifestations in patients suffering from defects in the urea cycle [15].
l-Glutamine + H_2_O → l-glutamate + ^+^NH_4_
(1)

Shortly after the discovery of PAG, Greenstein and Price discovered an apparent glutaminase activity in rat liver homogenates that was seemingly stimulated by pyruvate and other α-keto acids [16]. This activity was named **glutaminase II** [16] (***note the Roman numeral for two***). Meister and colleagues at the NIH subsequently showed that glutaminase II activity is actually due to a composite of two enzymes, namely an **l-glutamine transaminase** [Equation (2)] and an **ω-amidase** [Equation (3)] [17,18,19,20]. (Currently, transaminases are commonly referred to as aminotransferases; however, throughout this manuscript and with exceptions only where necessary, we use the term “transaminase” rather than aminotransferase to avoid confusion and retain the historical significance). The net glutaminase II reaction is shown in [Equation (4)]. ω-Amidase (ω-dicarboxylate amidohydrolase) was so named because it catalyzes the hydrolysis of the terminal amide (carboxamide) group of KGM (the α-keto acid analogue of l-glutamine; synonyms: 2-oxoglutaramate, 5-amino-2,5-dioxopentanoate), α-ketosuccinamate (KSM, the α-keto acid analogue of l-asparagine; synonyms: 2-oxosuccinamate, 4-amino-2,4-oxobutanoate), and several other monocarboxamides of 4- and 5-carbon dicarboxylates (but not that of l-glutamine or l-asparagine) [17,18,19,20]. The gene for ω-amidase is annotated as nitrilase-like protein 2 (*Nit2*) in the human genome.

In our past publications, we have used the term glutaminase II, as first used by Meister and colleagues, to describe the coupling of an l-glutamine transaminase with ω-amidase (e.g., [1,2,3]). However, we have become increasingly concerned that this term may have been confused with glutaminase 2 (GLS2), leading, in part, to the glutaminase II pathway being unappreciated by most researchers as having a distinct role in l-glutamine addiction in cancers. Therefore, to avoid confusion over nomenclature, we have recently renamed the glutaminase II pathway as the glutamine transaminase—ω-amidase (**GTωA** pathway) [4].
l-Glutamine + α-keto acid ⇆ α-ketoglutaramate (KGM) + l-amino acid (2)
KGM + H_2_O → α-ketoglutarate + ^+^NH_4_(3)
Net: l-Glutamine + α-keto acid + H_2_O → α-ketoglutarate + l-amino acid + ^+^NH_4_(4)

Meister and colleagues presented evidence that KGM is overwhelmingly present at neutral pH in the form of a lactam (2-hydroxy-5-oxoproline) (Figure 1) [19,20], a finding that was later verified by Hersh [21]. Hersh showed that, at equilibrium, in the physiological pH range, only 0.3% is in the open-chain form (the actual substrate of ω-amidase), whereas 99.7% is in the enzymatically inert lactam form. (Nevertheless, unless specified otherwise, the term KGM throughout the following text refers to the total (i.e., open-chain plus lactam) forms). Hersh also showed that the rate of interconversion between the two forms is base (i.e., ^−^OH)-catalyzed. At pH values above 8.5, the rate is so rapid that ring opening is not likely to be rate-determining for ω-amidase activity with KGM as substrate [21]. Thus, assays of ω-amidase activity with KGM as substrate are usually conducted at pH values of 8.5–9.0. (The enzyme has a broad pH optimum (~5–9) with KSM as substrate [19]). On the other hand, ring opening is slow enough at physiological pH values (~7.2) that despite inherently high activity of ω-amidase in many tissues (e.g., [22,23]), KGM is readily detected in tissues and body fluids. Thus, normal concentrations of KGM in rat tissues [24,25] and human cerebrospinal fluid (CSF) [26] are on the order of 1 μM to ~220 μM.

In conclusion, the glutaminase 1 (GLS1) and glutaminase 2 (GLS2) isozymes (and their splice variants) catalyze deamidation of l-glutamine to l-glutamate and ammonium in a ***single*** step [Equation (1)], whereas the formation of ammonium from the amide group of l-glutamine by the GTωA pathway requires ***two*** enzyme-catalyzed steps [Equations (2)–(4)].

## 3. Enzymes That Catalyze l-Glutamine Transamination and Subcellular Localization

Meister and colleagues partially purified an l-glutamine transaminase from rat liver and showed that the enzyme has broad specificity toward α-keto acids [17,18,19]. Subsequently, Cooper and Meister showed that rat tissues contain at least two l-glutamine transaminases [27,28]. Notably, these enzymes exhibited broad specificity toward both l-amino acids and α-keto acids. In view of their tissue predominance, one isozyme was named glutamine transaminase L (GTL), while the other isozyme, predominating in rat kidney, was named glutamine transaminase K (GTK) [28]. Subsequent studies of these enzymes demonstrated species differences in tissue content. Thus, mouse kidney has been reported to have a higher level of GTL than of GTK [29].

GTL and GTK activities were found to be present in both cytosolic and mitochondrial fractions of rat liver and kidney, respectively [28]. In this regard, the N-terminus of rat GTK possesses a 32 amino acid mitochondrial targeting sequence that allows entry into the mitochondria; removal of the sequence ensures a competing cytosolic location [30]. GTL (KAT3; see the following paragraph), which is annotated as KYAT3 in Uniprot (https://www.uniprot.org/uniprot/Q6YP21 (accessed on 22 July 2023)), is shown to exist as three isoforms generated by alternative splicing. Isoform 1 (Q6YP21-1; the canonical isoform) contains a mitochondrial targeting sequence. Isoform 3 (Q6YP21-3) lacks residues 1–34 of the canonical isoform. Isoforms 1 and 3 presumably account for the presence of the enzyme in the mitochondria and cytosol, respectively. In addition, a third isoform (Q6YP21-2) is listed, which is greatly truncated (missing residues 168–454) and contains an altered sequence 152–167. The function of this polypeptide is unknown. Thus, in summary, the presence of both GTK and GTL in mitochondria and cytosol is explained by alternative splicing of an N-terminal mitochondrial targeting sequence.

Owing to the broad substrate specificity of the l-glutamine transaminases, it is probable that some preparations of tyrosine, phenylalanine, serine, histidine, aminoadipate and kynurenine transaminases reported in the literature can be ascribed to l-glutamine transaminases—see the discussions in Refs. [2,22,31,32,33,34]. Of particular significance to the current discussion is the potential confusion over the nomenclature of the **kynurenine aminotransferases** (**KAT**s). Mammalian tissues exhibit at least four enzymes capable of catalyzing transamination of l-kynurenine (KAT1 through KAT4). KAT4 is identical to mitochondrial aspartate aminotransferase (AspAT), whereas KAT1 and KAT3 are identical to GTK and GTL, respectively [2,3,31,32,33,35,36,37,38,39,40]. The l-glutamine transaminase activities of KAT1 and KAT3 are discussed in more detail below (Section 5).

## 4. l-Glutamine Transaminases Catalyze l-Cysteine *S*-Conjugate β-Lyase (CCBL) and l-Selenocysteine *Se*-Conjugate β-Lyase Side Reactions

l-Cysteine *S*-conjugates [RSCH_2_CH(CO_2_^−^)(NH_3_^+^)] are intermediates in the mercapturate pathway for the detoxification of potentially toxic electrophiles. The l-cysteine *S*-conjugate formed with the electrophile is converted to the *N*-acetyl l-cysteine *S*-conjugate (mercapturate; [RSCH_2_CH(CO_2_^−^)(NHC(O)CH_3_)]) and excreted in the urine and/or bile ([41] and references cited therein). A further complication relating to nomenclature of the l-glutamine transaminases is the fact that many pyridoxal 5′-phosphate (PLP)-containing enzymes, including transaminases such as GTK/KAT1 and GTL/KAT3, catalyze β-lyase side reactions with cysteine *S*-conjugates, where RS^−^ is a good leaving group ([33,41,42] and references cited therein) [Equation (5)]. These side reactions compete with the formation of mercapturates. As a result of their cysteine *S*-conjugate β-lyase (CCBL) activity, GTK and GTL genes are listed in the human genome and by several commercial vendors with the alternative names *CCBL1* and *CCBL2*, respectively.
RSCH_2_CH(CO_2_^−^)(NH_3_^+^) + H_2_O → CH_3_C(O)CO_2_^−^ + ^+^NH_4_ + RSH (5)

It is particularly noteworthy that rat kidney GTK also catalyzes both transamination and competing β-lyase reactions with a number of l-selenocysteine *Se*-conjugates [RSeCH_2_CH(CO_2_^−^)(NH_3_^+^)] [33,42]. Both the transaminase and β-lyase activities are generally considerably greater with the selenium conjugates than with the corresponding sulfur conjugates [33,42]. An especially interesting selenocysteine *Se*-conjugate is *Se*-methyl-l-selenocysteine (MSC), a chemopreventive agent. MSC is both a transaminase and β-lyase substrate of GTK, although transamination is somewhat more favored over the β-lyase reaction [42,43,44]. It was suggested that part of the chemopreventive effect of MSC is due to β-elimination of methylselenol [CH_3_SeH] from MSC [45]. Transamination of MSC yields β-methylselenopyruvate [MSP; CH_3_SeCH_2_C(O)(CO_2_^−^)] [42]. Interestingly, MSP was shown to be a histone deacetylase (HDAC) inhibitor [43,44].

In a recent publication, Selvam et al. showed that introduction of high levels of GTK into hepatocellular carcinoma cell lines greatly enhanced the cytotoxicity of MSC, possibly as a result of increased oxidative stress [45]. The effect was even more pronounced in the presence of α-keto acid substrates of GTK/GTL, namely, phenylpyruvate and α-keto-γ-methiolbutyrate (KMB; synonyms: 4-methylthio-2-oxobutanoate, 4-methylthio-2-oxobutyrate) [45]. The presence of KMB presumably allowed maximal turnover of both transaminase and β-lyase activities of GTK by maintaining the active site coenzyme in the enzymatically active PLP form in the hepatocellular cells, as noted previously for purified human GTK [46]. Thus, production of MSP from MSC catalyzed by GTK/GTL may be chemoprotective to normal cells, but may be cytotoxic to cancer cells. Many electrophilic drugs are metabolized through the mercapturate pathway, including some anti-cancer drugs ([1,2,3,4] and references quoted therein). *In conclusion, any investigation of electrophilic drugs used in cancer therapy that are capable of forming cysteine S-conjugates with good leaving groups should take into account β-lyase reactions with cysteine S-conjugate metabolites and the possibility of a competing transamination reaction with endogenous l-glutamine*.

A generalized scheme for competing GTK/GTL-catalyzed transamination of l-cysteine *S*-conjugates versus β-elimination or *N*-acetylation is shown in Figure 2A. Figure 2B shows competing GTK/GTL-catalyzed transamination versus β elimination with the chemoprotectant *Se*-methyl-l-selenocysteine.

## 5. Relationship of l-Glutamine Transaminases to Kynurenine Aminotransferases (KATs)

Han et al. investigated a number of physiologically relevant amino acids as substrates of both human GTL/KAT3 and mouse GTK/KAT1 and showed that, among the amino acids tested as substrates, the highest catalytic efficiency (*V*_max_/*K*_m_) occurs with l-glutamine [35,36,37]. The values obtained with l-glutamine were notably greater than those obtained with l-kynurenine [35,36,37]. In other work, Han et al. showed that KAT2 (identical to aminoadipate aminotransferase and considered to be the main KAT in the brain) has some activity toward l-glutamine [39]. Moreover, l-glutamine is the most abundant amino acid in human tissues [47]. Its concentration is estimated to be 70–80 g per 70 kg individual, according to ([47] and references cited therein). Taking into account the molecular mass of l-glutamine [147.13 amu] and assuming 80% water content in the body, the average concentration of l-glutamine in human tissues is about 9 mM. On the other hand, the concentration of kynurenine in rat liver is reported to be 20 nmol/g (~22 μM), with much lower concentrations reported to be present in the brain, lung, and spleen [48]. *Thus, the **potential** rate of l-glutamine transamination catalyzed by GTK and GTL* in vivo *is likely to be orders of magnitude greater than that for l-kynurenine*. However, we cannot rule out the possibility that tissues are able to transaminate l-kynurenine to kynurenate in a compartment distinct from that catalyzing l-glutamine transamination, or that a relatively slow transamination of kynurenate by GTK/GTL is all that is required to maintain physiologically relevant concentrations of kynurenate in vivo. A perusal of PubMed (July 2023) showed more entries for the search term “kynurenine transaminase” (464) than for the search term “glutamine transaminase” (203). (Interestingly, several authors in the latter list misidentified the unrelated transglutaminases (TGases) as l-glutamine transaminases—yet another potential confusion in nomenclature! TGases catalyze the covalent cross-linking of the amide of an l-glutamine residue in a protein/peptide substrate to the ε-amino group of an l-lysine residue in a protein/peptide co-substrate (or to the amine group of smaller molecule co-substrates such as polyamines) and are not related to transaminases). The relatively high citation rate for KATs is presumably in part because the product of l-kynurenine transamination (i.e., kynurenate) has been extensively studied as a neuromodulator/neuroprotectant against l-glutamate excitotoxicity in normal and disease states (e.g., [49] and references cited therein).

l-Kynurenine is formed from l-tryptophan by the action of indoleamine 2,3-dioxygenases 1 and 2 or by tryptophan 2,3-dioxygenase [49]. In addition to transamination to kynurenate catalyzed by the various KATs, l-kynurenine is converted to 3-hydroxyanthranilate by the action of kynureninase (also known as l-kynurenine hydrolase). 3-Hydroxyanthranilate is a precursor of NAD^+^ via quinolinate [49]. Considerable effort has been directed toward the development of KAT inhibitors [50,51] as possible therapeutic agents, mostly directed toward neurological disorders. We emphasize these points because there is also a wealth of literature regarding l-kynurenine and cancer. A PubMed search for the terms “cancer” and “kynurenine” yielded 1303 entries (July 2023). As far as we can discern, most of these entries are related *to indoleamine 2,3-dioxygenases and not to KATs. Nevertheless, researchers interested in the role of kynurenine in cancer should be aware of the broad specificity of KATs, especially in their roles as l-glutamine transaminases and as cysteine S-conjugate β-lyases*.

## 6. Evidence That Enzymes of GTωA Pathway Operate Extensively In Vivo

### 6.1. ω-Amidase Is Widespread in Mammalian Tissues

When measured under optimal pH conditions (pH 8.5), ω-amidase activity (with KGM as substrate) was shown to be present in all ten rat tissues investigated, with the highest specific activity in liver and kidney [22]. This finding is in agreement with a previous study by Meister, who showed that ω-amidase activity is present in all eight rat tissues investigated, with the highest specific activity in liver and kidney [19]. These findings regarding the specific activity of ω-amidase in rat tissues may also be compared to findings about human Nit2 (i.e., ω-amidase) mRNA [52]. Lin et al. showed that the mRNA for Nit2 is well represented in all 16 human tissues/cell types they investigated, with the highest levels present in liver and kidney [52]. When measured under optimal conditions at near-saturation levels of KGM (≥5 mM), 25 °C or 37 °C, and high pH (8.5–9.0), the specific activity of the purified rat, mouse, and human liver enzymes is relatively high (~5–50 μmol/min/mg protein) [53,54,55,56,57]. The *K*_m_ values for KGM exhibited by human and mouse ω-amidase have been reported to be ~2.7 and ~0.6 μM, respectively (pH 8.5; 30 °C), when corrected for the amount of open-chain form present in solution [54,56]. (At a concentration of 5 mM, ~15 μM will be in the open-chain, substrate form). Thus, the enzyme has relatively high affinity for open-chain KGM, and ***inherent*** activity of ω-amidase with KGM is high in rat, mouse, and human tissues.

ω-Amidase activity is present in both the cytosolic and mitochondrial fractions of rat liver and kidney [22,23] and presumably in these fractions in other tissues. The mechanism by which ω-amidase is imported into the mitochondria is currently under investigation.

### 6.2. High Inherent l-Glutamine Transaminase Activities in Mammalian/Human Tissues

With the use of appropriate α-keto acid co-substrates, it has been shown that the inherent capacity for l-glutamine transaminase activity is relatively high in rat kidney and liver [22,23,28]. Moderate to high activity is also present in a variety of other rat tissues. In these studies (i.e., [22,23,28]), rat prostate was not examined. The occurrences of l-glutamine transaminase and ω-amidase activities in normal rat prostate and in normal and cancerous human prostate are discussed separately below (Section 12.2).

The capacity of isolated, perfused rat liver to transaminate l-glutamine is very high, as found in [58]. In that study, the authors showed that phenylpyruvate and KMB (which, as noted above, are substrates of l-glutamine transaminases) added to the perfusate were effectively transaminated by the liver using endogenous and perfusate l-glutamine as amine donor. Under certain conditions, the capacity of the GTωA pathway exceeded that of PAG in the presence of these α-keto acids. The amide nitrogen of endogenous and perfusate l-glutamine was quantitatively recovered as l-glutamate, ammonium, urea, and l-alanine [58]. These results suggest that the KGM formed within the hepatic acinus was effectively deamidated and that the ammonium thus formed was incorporated into urea or l-glutamate; the l-glutamate was then transaminated with pyruvate, as a result of high endogenous activity of hepatic alanine aminotransferase (AlaAT) [58]. Moreover, α-cyano-4-hydroxycinnamate, an inhibitor of the mitochondrial membrane monocarboxylate translocators (transporters) 1–4 (MCTs 1–4), inhibited l-glutamine uptake and the release of l-methionine (formed from transamination of KMB) by about 30% in the presence of added KMB [58]. The authors proposed that this finding might suggest that about 2/3 of l-glutamine transamination is cytosolic [58]. As a corollary, this finding also suggests that about 1/3 of l-glutamine transamination in rat liver may be mitochondrial, an important point that is consistent with the presence of ω-amidase activity in these organelles. The importance of the GTωA pathway in mitochondria and its importance to cancer biology will be discussed later.

Human GTK/KAT1 and mouse GTL/KAT3 exhibit apparent *K*_m_ values of 2.8 mM [36] and 0.7 mM [37], respectively, toward l-glutamine. On the other hand, Botman et al. reported that mouse kidney and brain tissue PAG exhibit *K*_m_ values for l-glutamine of 0.6 mM whereas mouse liver exhibits a *K*_m_ value for l-glutamine of 11.6 mM [59]. These findings suggest that the l-glutamine transaminases possess similar affinities for endogenous l-glutamine as do GLS1 and GLS2. However, the findings do not take into account relative *V*_max_ values. Nevertheless, an important but overlooked study by Darmaun et al. [60] suggests that glutamine transaminase activity in vivo is considerable. These authors studied the turnover of l-[^15^N]glutamate, l-[2-^15^N]glutamine, and l-[5-^15^N]glutamine in adult volunteers and suggested that whole-body ammonium production via the GTωA pathway [Equation (4)] is substantial and may even exceed that of the PAG pathway [Equation (1)]. However, the authors raised one caveat: the reversibility of most transaminase reactions may have complicated interpretation of the results [60]. Nevertheless, because the product of l-glutamine transamination (i.e., KGM) exists overwhelmingly in the lactam form and because of the presence of ω-amidase, the GTωA pathway irreversibly favors formation of α-ketoglutarate and ammonium from l-glutamine [Equation (4)]. *Thus, the work of Darmaun* [60] *is very strong evidence that the rate of turnover of l-glutamine* via *the GTωA pathway in humans is substantial*.

### 6.3. Occurrence of KGM in Rat Tissues and Clinical Samples

Despite inherent high activity of tissue ω-amidase, as noted above, the concentration of KGM in liver and kidney is of the order of low μM to ~220 μM [24,25], presumably as a result of the relatively slow rate of ring opening of the KGM lactam to the enzymatically active open-chain form at pH values ~7.2. Nevertheless, the presence of KGM in rat tissues is very strong *a priori* evidence that transamination of l-glutamine occurs in vivo. In addition to its occurrence in rat tissues, KGM, as noted above, is present in human CSF [26]. Interestingly, the concentration of KGM is increased in the CSF of hyperammonemic patients with hepatic encephalopathy, possibly due in part to elevated concentrations of tissue l-glutamine [26,61]. The urinary concentration of KGM is also increased in patients with primary hyperammonemia who exhibit a defect of the urea cycle, possibly in part also as a result of elevated l-glutamine concentrations [62,63]. In addition, the concentration of urinary KGM is increased in patients with secondary hyperammonemia due to a deficiency of citrin (a mitochondrial glutamate/aspartate transporter) [64]. Thus, KGM appears to be a good biomarker for several diseases characterized by hyperammonemia. There is considerable evidence that excess circulating ammonium is neurotoxic, but there is substantial debate concerning the mechanism of this toxicity (e.g., [65] and references quoted therein). Neurotoxicity may be due to (1) production of excess toxins, (2) energy imbalance, (3) pro-inflammatory processes, and (4) central neurotransmission imbalance in favor of excess neuronal inhibition mediated by GABA [65]. An additional underlying factor contributing to hyperammonemia-induced neurotoxicity is the increased l-glutamine concentration in astrocytes (the cell type where most of the brain glutamine synthetase is located). This leads to osmotic stress, swelling, and damage to the brain—the osmotic gliopathy theory [66].

Hyperammonemia occurs in many cancers and/or during treatment, and this contributes to poor prognosis (e.g., see [67,68,69] for some recent references). *It would be interesting to determine whether blood and/or urine KGM is a biomarker for certain cancers, metastases and/or treatments in which hyperammonemia is a confounding feature*. Studies have shown that lowering ammonium levels can increase responses of tumors to treatment [70].

In a recent study, Udupa et al. [71] showed that treatment of patient-derived pancreatic orthotopic JHU094 tumors in male Foxn1^nu^ athymic nude mice with BPTES (bis-2-(5-phenylacetamido-1,3,4-thiadiazol-2-yl)ethyl sulfide) encapsulated in nanoparticles (BPTES-NP) resulted in a large increase in endogenous KGM. (JHU is an acronym for Johns Hopkins University; BPTES is an inhibitor of allosteric binding of phosphate to GLS1/GAC [72,73,74,75,76]). The increase is presumably due, at least in part, to the increased levels of l-glutamine available for transamination.

### 6.4. Possible Involvement of KGM in Urea Nitrogen Formation and in Acid–Base Balance

As a result of incomplete hydrolysis of KGM in extrarenal and extrahepatic tissues, KGM accumulates in the blood. As noted above, the concentration in rat blood is ~19 μM. We suggest that once in the circulation, a portion of this KGM is taken up by the liver, where it enters a larger pool of KGM (~216 μM in rat liver [25]). This pool of KGM can then theoretically act as a source of ammonium for urea synthesis, for glutamine synthesis, or for both (Figure 3). Thus, in future studies, it will be important to determine whether ω-amidase is present in the urea-cycle-containing periportal hepatocytes, the glutamine-synthetase-containing perivenous hepatocytes, or both. In addition, it is possible that a portion of the blood-borne KGM is also taken up by the kidneys, where it can be excreted as ammonium ion as part of acid–base regulation by the kidneys; the possible involvement of the GTωA pathway in this process is briefly discussed in reference [3]. It is known that, in the rat kidney, GTK is confined largely to the proximal tubules—see references quoted in [3]. In future work, it will be important to determine the cellular location of ω-amidase in rat kidney and its location relative to GTK. *We hypothesize that, notwithstanding the lack of current knowledge regarding the exact cellular location of ω-amidase in the liver and kidney, KGM is an important “player” in the maintenance of nitrogen homeostasis in mammals*.

## 7. Additional Proposed Biological Roles of the GTωA Pathway

### 7.1. Closure of the Methionine Salvage Pathway by Transamination of KMB

A major route for l-methionine metabolism involves its conversion to *S*-adenosyl-l-methionine (SAM). SAM is the major methyl donor in vivo. However, a substantial portion of SAM is also involved in polyamine biosynthesis. During polyamine biosynthesis, carbon 1 of l-methionine is converted to CO_2_, and carbons 2–4 and the amine nitrogen are incorporated into polyamines, whereas the sulfur and methyl group are incorporated into 5′-methythioadenosine (MTA). In order to prevent a drain of potentially scarce sulfur and methyl moieties, a salvage pathway arose early in evolution that is widespread in nature, occurring in unicellular bacteria, archaea, fungi, and in mammalian and plant cells (e.g., [77,78]). This pathway converts MTA to l-methionine. During this process, the original sulfur and methyl moiety are retained in the newly formed l-methionine, whereas carbons 1–4 are obtained anew from the ribose portion of MTA. The intermediates in the salvage pathway were brilliantly elucidated by Abeles and colleagues at Brandeis University in the 1990s [79,80,81]. Notably, the last step of the pathway is closed by transamination of KMB to l-methionine with a suitable amino acid. It has been known for over 40 years that, in rats, transamination of KMB occurs with l-glutamine and, to a lesser extent, with l-asparagine [82]. Moreover, an l-glutamine transaminase homologue and ω-amidase potentially act in tandem to close the methionine salvage pathway in bacteria and plants [83].

Substituting KMB for α-keto acid in Equation (4) yields Equation (6) (closure of the methionine salvage pathway). Thus, closure of the methionine salvage pathway, by means of the GTωA pathway, has the advantage that not only is l-methionine salvaged, but simultaneously, anaplerotic α-ketoglutarate is produced from expendable l-glutamine.
l-Glutamine + α-keto-γ-methiolbutyrate (KMB) + H_2_O → α-ketoglutarate + l-methionine + ^+^NH_4_
(6)

Inhibitors of enzymes within the methionine salvage pathway have been considered as anti-cancer agents. For example, methylthio-DADMe-immucillin-A ((3R,4S)-1-[(4-amino-5H-pyrrolo [3,2-d]pyrimidin-7-yl)methyl]-4-[(methylthio)methyl]-3-pyrrolidinol, MT-DADMe-ImmA), a potent transition state inhibitor (*K_i_* = 1.7 nM) analogue of 5′-methylthioadenosine phosphorylase (MTAP; the rate-limiting enzyme of the l-methionine salvage pathway), possesses anti-cancer properties in several in vitro cancer cell lines and in in vivo mouse models of human cancers [84,85]. Mouse xenografts of FaDu head and neck cancer and A549 and H358 lung cancers in immunocompromised mice are susceptible to oral treatment with MTDIA [84,85]. MT-DADMe-ImmA was also shown to extend lifespan in an APC^Min/+^ mouse model of human familial adenomatous polyposis coli, in which patients are at increased risk for colorectal cancer [86]. In another example of an anti-cancer effect resulting from disruption of l-methionine salvage, Affronti et al. showed that inhibition of MTAP, together with upregulation of spermidine/spermine N^1^-acetyltransferase (SSAT) activity, resulted in increased apoptosis in radical prostatectomy ex vivo explant prostate cancer samples [87]. The authors suggested that increasing l-methionine-derived carbon flux through polyamine biosynthesis (i.e., increasing SSAT activity) while simultaneously inhibiting MTAP “results in an effective therapeutic approach potentially translatable to the clinic” [87]. *We suggest that inhibitors of the GTωA pathway, which will interfere with closure of the last step of the methionine salvage pathway, may exhibit synergistic or added efficacy with other anti-glutaminase pathway drugs against prostate cancer and other cancers*.

### 7.2. Salvage/Detoxification of α-Keto Acids

l-Amino acid [RCH(CO_2_^−^)(NH_3_^+^)] and α-keto acid [RC(O)CO_2_^−^] substrates of GTK and GTL generally possess an R group that is hydrophobic or uncharged—e.g., l-glutamine, l-phenylalanine, l-tyrosine, l-methionine, l-kynurenine, phenylpyruvate, *p*-hydroxyphenylpyruvate, and KMB. However, although l-leucine is a good substrate of human GTK/KAT1, l-valine, l-isoleucine, and the corresponding α-keto acids are poor substrates [36], presumably due to branching at the β carbon, which hinders binding at the active site. l-Valine, l-isoleucine, and the corresponding α-keto acids are also poor substrates of mouse GTL/KAT3 [37]. Curiously, l-leucine is also a poor substrate of GTL/KAT3, but its corresponding α-keto acid (α-ketoisocaproate) is a moderately good substrate [37]. l-Glutamate, l-aspartate, and their corresponding α-keto acids are very poor substrates of both GTK and GTL [36,37], presumably as a result of the charged R group. This makes biological sense—*if the α-keto acid in the general equation for the GTωA pathway [Equation (4)] were α-ketoglutarate, then the net reaction would be the conversion of l-glutamine to l-glutamate and ammonium with **no** net production of anaplerotic α-ketoglutarate from l-glutamine*.

The aldehyde glyoxylate [HC(O)CO_2_^−^; R = H], but not the corresponding amino acid (glycine), is a good substrate of both GTK and GTL, presumably because it can fit easily into the active site. Failure to convert glyoxylate to glycine by an aminotransferase (peroxisomal alanine aminotransferase) mistargeted to mitochondria results in hyperoxaluria type 1 and potentially the formation of renal calculi (calcium oxalate stones) ([88] and references cited therein). The l-glutamine transaminases may play a role in preventing the accumulation of potentially toxic oxalate in the kidneys by removing the precursor, glyoxylate [40].

Many transaminases, such as AspAT and alanine aminotransferase (AlaAT), exhibit less than “perfect” specificity. For example, both the cytosolic and mitochondrial isozymes of AspAT exhibit some activity toward l-phenylalanine and l-tyrosine [89,90]. We have hypothesized that one role of the GTωA pathway is to convert α-keto acids arising through non-specific transamination reactions back to the corresponding l-amino acid [2,3,22,28,32]. Thus, the GTωA pathway may act to repair non-canonical pathways of other transaminases. Evidence for this hypothesis has recently been published [91]. Inspection of Equation (4) shows that the pathway ensures ***irreversible*** removal of α-keto acids at the expense of readily obtainable l-glutamine. Some of these α-keto acids may be toxic at higher concentrations (e.g., phenylpyruvate, α-ketoisocaproate (αKIC), glyoxylate). At the same time, the carbon skeleton of l-glutamine is not “wasted” but is converted to anaplerotically useful α-ketoglutarate. The literature is replete with studies demonstrating amino acid requirements of cancer cells and their microenvironment. For example, several recent reviews emphasize the role of amino acids in promoting protein synthesis, maintaining biosynthetic processes, acting as an energy source, and maintaining redox balance in cancer cells (e.g., [92,93,94]). For many of the common protein amino acids, the first step in their metabolism involves conversion to the corresponding α-keto acid (see the next section). *We suggest that the GTωA pathway regulates the concentrations of certain amino acids and α-keto acids in order to balance protein synthesis in rapidly dividing cancer cells with energy production from the metabolism of these compounds*.

### 7.3. Possible Anti-Oxidant Role of the GTωA Pathway and Its Response to Hypoxia

It has been known for over 80 years that typical α-keto acids are oxidized quantitatively to carboxylates by hydrogen peroxide [RC(O)CO_2_^−^ + H_2_O_2_ → RCO_2_^−^ + CO_2_ + H_2_O] ([95] references quoted therein). Indeed, it has been suggested that α-ketoglutarate [^−^O_2_CCH_2_CH_2_C(O)CO_2_^−^] can function directly as an anti-oxidant in vivo by reacting with H_2_O_2_ to form succinate [^−^O_2_CCH_2_CH_2_CO_2_^−^], H_2_O, and CO_2_ (e.g., [96]). Succinate can then be metabolized through the TCA cycle [96]. In addition, α-ketoglutarate has been suggested to stimulate anti-oxidative defenses [97]. Finally, α-ketoglutarate (2-oxoglutarate; 2-OG) is known to play a role in the regulation of the transcription factor HIF-1α. Under normoxic conditions, oxygen-dependent prolyl hydroxylases [EGLN1-3; α-ketoglutarate/2-oxoglutarate-dependent hydroxylases (2OGDDs)] in the presence of α-ketoglutarate and O_2_ hydroxylate HIF-1α, marking it for ubiquitination and rapid degradation by the proteasome. Under low oxygen concentrations, this process is impeded, allowing stabilization of HIF-1α. HIF-1α forms a heterodimer with HIF-1β, which then translocates to the nucleus, where it binds to hypoxia response elements (HREs) to release promoter-paused RNApol2 and enhance gene transcription [98]. For a recent review on the role of 2OGDDs in normal cells and how they are “corrupted” in cancer, see [99]. Whether α-ketoglutarate generated by the GTωA pathway is a substrate for the 2OGDDs remains to be determined. In conclusion, from several perspectives, the GTωA pathway, as a source of α-ketoglutarate, may play a role in anti-oxidant defenses and upregulation of defense systems under hypoxic conditions. The role of the GTωA pathway in these processes needs to be investigated but will not be discussed further here.

## 8. Possible Role of the GTωA Pathway in Transferring α-Keto Acid/l-Amino Acid Carbon between Cellular and Subcellular Compartments

Many amino acids, including the branched-chain amino acids (BCAAs), l-aspartate, l-alanine, and l-tyrosine, are metabolized through pathways in which the major first step is linked to transamination with α-ketoglutarate. We have suggested that linkage of α-ketoglutarate-dependent transaminases [Equation (7)] with the GTωA pathway [Equation (4)] affords a metabolically gainful mechanism for transportation of certain l-amino acids between two compartments [2,3,4]. In this case, movement of an l-amino acid from compartment 1 to compartment 2 is associated with movement of the corresponding α-keto acid in the reverse direction. Loss of α-keto acid in compartment 1 will be accompanied by appearance of this α-keto acid in compartment 2. The compartments may be at the cellular level or the subcellular level. Indeed, as mentioned above, the GTωA pathway is widely distributed in mammalian tissues and is present in ***both*** cytosolic and mitochondrial fractions ([2] and references quoted therein). This contrasts with GLS1 and GLS2, which, as noted above, are mitochondrial enzymes and do not participate directly in the transfer of α-keto acids and l-amino acids between cytosol and mitochondria.

MCTs 1–4 are well-characterized proton-dependent transporters of glycolytic products (i.e., lactate and pyruvate) as well as ketone bodies (acetoacetate and β-hydroxybutyrate) across the plasma membrane. MCTs 1, 2, and 4 (l-lactate transporters) are overexpressed in many cancers. For example, MCT1 is overexpressed in non-small-cell lung cancers and may be a prognostic marker for survival [100]. *Thus, the versatility of the GTωA pathway, together with suitable membrane transporters such as MCTs 1, 2, and 4, which are overexpressed in many cancers, provides a metabolic network to facilitate and channel amino acids and α-keto acids to appropriate organelles and compartments critical for energy production or biosynthetic pathways*. Inhibitors of MCTs are being considered as therapeutic targets in cancer [101]. *We suggest that inhibitors of the GTωA pathway may be useful adjuncts to such inhibitors.*
**Compartment 1: l-Amino acid** + α-ketoglutarate ⇆ α-keto acid + l-glutamate(7)
**Compartment 2:** l-Glutamine + α-keto acid + H_2_O → α-ketoglutarate + **l-amino acid** + ^+^NH_4_
(4)

## 9. “Glutamine Addiction” and the Canonical Pathway for α-Ketoglutarate Formation in Cancer Cells

### 9.1. The Canonical Pathway for Satisfying “Glutamine Addiction” in Cancer Cells

Many cancer cells use glycolysis to generate lactate from circulating glucose even under normoxic conditions (the Warburg effect) despite the fact that these cells possess a functioning TCA cycle. For these cancer cells, the TCA cycle is used mainly to provide anaplerotic carbon necessary for major energy requirements and for building biomass in rapidly dividing cells. l-Glutamine is a major source of this anaplerotic carbon via its conversion to α-ketoglutarate, while, at the same time, the nitrogen is used for DNA synthesis (e.g., [102,103,104,105,106,107,108,109]). As noted above, this process is commonly referred to as “glutamine addiction”. In almost all studies of glutamine addiction in cancer cells, it is assumed that l-glutamine is converted to α-ketoglutarate via a pathway that is initiated by the action of a glutaminase [Equation (1)]. The l-glutamate thus formed is then converted to α-ketoglutarate by an l-glutamate-linked transaminase [Equations (7) and (8) shown in the reverse direction)] or via glutamate dehydrogenase (GDH) [Equation (9)]. The flow of carbon may be depicted as Gln → Glu ⇆ α-ketoglutarate. It has been suggested that, in the case of hepatocellular carcinoma, the transamination step predominates in the presence of adequate glucose, whereas the GDH step predominates at low glucose concentrations [110]. A key feature of mammalian GDH (GDH1 in higher primates) is that it is heavily allosterically regulated by an array of activators, e.g., ADP and l-leucine, and inhibitors, e.g., GTP, palmitoyl CoA, and ATP [111]. The NAD(P)^+^/NAD(P)H ratio has been reported to be five times higher in cancerous human colon cells compared to normal human colon cells [112]. This would favor oxidative deamination of l-glutamate (i.e., the forward direction of Equation (9)). Nevertheless, the GTωA pathway would be energetically more favorable for the conversion of l-glutamine to α-ketoglutarate than the GLS/GDH pathway in cancer biology (Equation (4) is irreversible, whereas Equation (9) is reversible). The role of GDH in cancers is discussed below.
**l-Glutamine** + H_2_O → l-glutamate + ^+^NH_4_ **Deamidation**(1)
l-Glutamate + α-keto acid ⇆ **α-ketoglutarate** + l-amino acid **Transamination**(8)
l-Glutamate + NAD(P)^+^ + H_2_O ⇆ **α-ketoglutarate** + ^+^NH_4_ + NAD(P)H (9)

It is important to note that directionality of the above GDH reaction is linked to redox homeostasis of the cell, in particular to the ratio of its pyridine nucleotide coenzymes, NAD^+^/NADH and NADP^+^/NADPH. The NAD^+^-dependent GDH functions mainly in a metabolic capacity: i.e., NAD^+^ is used when catabolizing l-glutamate to form anaplerotic α-ketoglutarate, ammonium, and NADH. In addition, the l-glutamate to α-ketoglutarate reaction is coupled to energy demands, being activated by high levels of ADP and GDP, as well as by low blood glucose levels. In contrast, the NADPH-specific enzyme in response to elevated ATP conducts a biosynthetic reaction that converts ammonium and α-ketoglutarate to l-glutamate [111].

Most studies of l-glutamine addiction in cancer cells have consistently shown a pro-oncogenic role for GLS1 as a source of l-glutamate destined to be converted to α-ketoglutarate. However, the situation with GLS2 is not as straightforward. There is uncertainty as to whether GLS2 is predominantly a tumor suppressor or tumor promoter in some cases (e.g., [113]). Nevertheless, strong evidence suggests that GLS2 is a tumor suppressor in hepatic carcinoma [114,115]. *Our studies suggest that, regardless of the exact role of GLS2 in cancers, cancer researchers interested in studying glutamine addiction in mitotic diseases (including cancers *[116]*) must take into account the presence of the GTωA pathway in these cells*.

### 9.2. What Are the Transaminases (Aminotransferases) That Can Potentially Supply α-Ketoglutarate to the TCA Cycle in the Canonical Pathway [Equation (8)]?

We consider three major transaminases here and discuss the probabilities of their participation in the canonical l-glutamine to α-ketoglutarate pathway.

#### 9.2.1. Alanine Aminotransferase (AlaAT; Glutamate Pyruvate Transaminase (GPT))

Most tissues contain AlaAT activity [Equation (10)], but the enzyme has its highest specific activity in the liver. In the liver, the enzyme ensures that l-alanine is highly gluconeogenic: i.e., formation of pyruvate is favored via the reverse direction of Equation (10). For example, it has been known for over 50 years that, of the 20 common protein amino acids, l-alanine is the most abundantly released by human muscle and that this amino acid is the major amino acid extracted by the liver for gluconeogenesis; glucose is released to the circulation, where it provides a source of pyruvate for l-alanine production by the muscle completing an l-alanine cycle [117].

Zebrafish have recently been suggested to be a new model for investigating crosstalk between host and tumor metabolism [118]. Interestingly, a recent study shows an l-alanine cycle between a melanoma and liver in zebrafish [119]. The melanoma was found to consume a relatively large amount of glucose and, at the same time, excrete l-alanine [119]. The authors suggested that excretion of glucose-derived l-alanine from tumors provides a source of carbon for hepatic gluconeogenesis and allows tumors to remove excess nitrogen from BCAA catabolism, which was found to be activated in zebrafish and human melanomas [119]. Whether this cancer–tissue axis is a generalized finding for other tumors remains to be elucidated. Nevertheless, in this zebrafish model of a human cancer, the findings suggest that considerable transamination of l-glutamate with pyruvate must have occurred to generate a large, releasable pool of l-alanine from the tumor. At first glance, this finding would seem to be in accord with the importance of a transamination reaction (in this case AlaAT) for generating α-ketoglutarate from l-glutamine via the canonical route. However, in order for the nitrogen of BCAAs to be incorporated into l-alanine, an α-ketoglutarate linked BCAA aminotransferase (BCAT) [Equation (11)] must be linked to AlaAT. The net reaction is shown in Equation (12), where it becomes evident that there is ***no***
*net generation or removal of α-ketoglutarate*. Similar arguments hold for alanine metabolism in mammalian tissues.
l-Glutamate + pyruvate ⇆ α-ketoglutarate + l-alanine (10)
α-Ketoglutarate + l-branched-chain amino acid ⇆ l-glutamate + branched-chain α-keto acid(11)
Net: Pyruvate + l-branched-chain amino acid ⇆ l-alanine + branched-chain α-keto acid (12)

#### 9.2.2. Aspartate Aminotransferase (AspAT; Glutamate Oxaloacetate Transaminase (GOT))

AspAT [Equation (13)] is very active in most tissues and occurs as two isozymes—a cytosolic and a mitochondrial isozyme. AspAT is an important enzyme in cancer cell intermediary metabolism [120,121]. (The title of Ref. [120] uses the wording “l-glutamine-utilizing transaminases”, but this is clearly a misnomer). In that study, both AspAT and phosphoserine aminotransferase were shown to be important sources of α-ketoglutarate in breast cancer cells [120]. Inhibitors of AspAT have been considered to be potential anti-cancer agents (e.g., [121]). It should be noted that both oxaloacetate and α-ketoglutarate are important components of the TCA cycle. Therefore, inflow of carbon into the TCA cycle as α-ketoglutarate is accompanied by conversion of this carbon to oxaloacetate and CO_2_. This is formally depicted in [Equation (14)], where only one carbon [-CH_2_-] moiety is oxidized through the TCA cycle. This process will generate 1 GTP and ~8 ATPs per -CH_2_- moiety oxidized. However, the mitochondrial membrane is impervious to oxaloacetate [122]. Thus, oxaloacetate in the cytosol must be generated in a manner independent of the TCA cycle. This can be accomplished by oxidation of malate via the malate dehydrogenase (MDH) reaction [Equation (15)].
l-Glutamate + oxaloacetate ⇆ α-ketoglutarate + l-aspartate (13)
α-Ketoglutarate + [O] → oxaloacetate + H_2_O + CO_2_
(14)
Malate + NAD^+^ ⇆ oxaloacetate + NADH + H^+^
(15)

It has long been known that mitochondria are also impervious to NADH generated by glycolysis in the cytosol. Thus, electrons associated with NADH generated in the cytosol must be shunted (shuttled) from the cytosol into the mitochondria to couple glycolysis to oxidative phosphorylation and the electron transport chain. A major shuttle for the transfer of electrons, in lieu of NADH, into the mitochondrial electron transport chain is the malate–aspartate shuttle (MAS; also known as the Borst cycle), which requires the participation of both cytosolic and mitochondrial isozymes of AspAT, mitochondrial and cytosolic isozymes of MDH, an l-aspartate/l-glutamate mitochondrial membrane transporter, and a malate/α-ketoglutarate mitochondrial transporter, according to ([122] and references cited therein). For a review of the MAS (and the glycerol phosphate shuttle) for the transfer of reducing equivalents between cytosol and mitochondria in the brain, see Ref. [123]. During operation of the MAS, malate is taken up into the mitochondria on a transporter that extrudes α-ketoglutarate while l-glutamate is simultaneously taken up by the mitochondria and l-aspartate is expelled. One result of the MAS shuttle is the net transamination of l-glutamate with oxaloacetate to α-ketoglutarate and l-aspartate in the mitochondria and net transamination of α-ketoglutarate with l-aspartate to l-glutamate and oxaloacetate in the cytosol. In the absence or reduced capacity of the shuttle, the electrons associated with the NADH generated in the cytosol are not transferred to the mitochondria via reduction of oxaloacetate to malate and instead are used to reduce pyruvate to lactate. As noted above, many cancer cells generate lactate from glucose despite the presence of a functional TCA cycle in the mitochondria. Even though this process is inefficient at generating ATP from the metabolism of glucose, many cancer cells rely on “profligate” use of circulating glucose and, at the same time, release copious amounts of lactate. However, the extent of uncoupling of glycolysis from the TCA cycle may fluctuate depending on phenotypic variations in the tumor. For example, 80% of the total ATP production in melanoma and lung cancer cells and approximately 40% of the ATP production in pancreatic cancer cells have been estimated to rely on the MAS for NADH transport into the mitochondrial matrix, according to ([124] and references cited therein).

#### 9.2.3. Branched-Chain Amino Acid (BCAA) Aminotransferases (BCATs)

Mammalian tissues contain two BCAT isozymes, namely a cytosolic isozyme (BCATc (BCAT1)) and a mitochondrial isozyme (BCATm (BCAT2)) [125]. The α-keto acid acceptor of these enzymes is α-ketoglutarate [Equation (11)], and the enzymes play a key role in the catabolism of the branched-chain amino acids [126,127]. Indeed, there is evidence that BCAT1 is part of a supramolecular complex (metabolon) with the next enzyme system in the branched-chain catabolic pathway (i.e., the branched-chain α-keto acid dehydrogenase complex) [124]. The brain is especially rich in both BCAT isozymes, and it has been proposed that these isozymes play an important role in a BCAA-dependent glial–neuronal nitrogen shuttle [128]. Nevertheless, under normal conditions, the BCATs are a net source of l-glutamate and not of α-ketoglutarate [125,128].

A considerable number of studies cite the importance of BCAAs in nitrogen transfer within and between cells and tissues and demonstrate the metabolic interplay in protein and energy metabolism and nutrient signaling in mammalian targets of rapamycin (mTOR) activity, (e.g., [129]). The possible role of BCAAs in cancer cells is discussed in more detail in Section 13.

#### 9.2.4. Conclusions Regarding Contribution of l-Glutamate-Utilizing Transaminases to the Canonical Pathway

Two major transaminases (AlaAT, BCAT) are likely to catalyze the net conversion of their l-amino acid substrates to the corresponding α-keto acids in cancer. This process will convert α-ketoglutarate to l-glutamate rather than vice versa. Thus, we suggest that these transaminases may not be ***net*** contributors for the conversion of l-glutamate to α-ketoglutarate in cancer cells. Nevertheless, AspAT appears to be a good candidate for production of α-ketoglutarate from l-glutamate in glutamine addiction, and as noted above, both isozymes are extremely active in mammalian tissues. Indeed, the components of the reaction are thought to be near thermodynamic equilibrium in liver [130] and brain [131,132]. Accordingly, l-glutamate generated from l-glutamine will rapidly equilibrate with the l-glutamate pool available to the AspAT isozymes. The components of this equilibrium, in turn, are shared with components of a number of enzyme systems/metabolic pathways, transporters, and the MAS. For example, α-ketoglutarate and l-glutamate are intermediates in the metabolism of not only l-glutamine but also, e.g., l-proline, l-arginine, and l-ornithine. Thus, the flow of α-ketoglutarate generated from l-glutamate via AspAT is unlikely to be directed stoichiometrically into the TCA cycle. These considerations should apply to both normal and cancerous tissues.

In conclusion, in the canonical pathway for glutamine addiction, α-ketoglutarate is formed from l-glutamate by **reversible** transamination (or by the **reversible** GDH reaction, which is also dependent on the NAD(P)^+^/NAD(P)H ratios of the cell; see the next section). On the other hand, the formation of α-ketoglutarate by the GTωA pathway is **irreversible**. Thus, the GTωA pathway permits a more efficient coupling of α-ketoglutarate to the TCA cycle than that permitted by the canonical pathway. Moreover, as noted above, the GTωA pathway occurs in mitochondria. A search for mitochondrial-binding partners to ω-amidase that can utilize α-ketoglutarate generated by ω-amidase is currently under way.

### 9.3. Possible Role of Glutamate Dehydrogenase (GDH) in the Production of α-Ketoglutarate in the Canonical Pathway

The reaction catalyzed by GDH is shown in Equation (9). Whereas most mammals possess a single GDH isozyme (GDH1), higher primates, including humans, possess a second isozyme (GDH2) that apparently arose through duplication of the *GLUD1* gene followed by rapid evolutionary drift in parallel with the evolution of increased brain size [133]. In humans, the intron-containing *hGLUD1* gene is on chromosome 10, whereas the intronless *hGLUD2* gene arose through retropositioning of the *hGLUD1* gene to chromosome X [133]. In humans, GDH1 is expressed in most organs but has its highest specific activity in liver. GDH2 is highly expressed in brain, kidney, and testes (and other steroidogenic organs). In contrast, it is absent from liver [133]. Both isozymes are mostly located in mitochondria in the cell, and both are activated by l-leucine and ADP. However, only hGDH1 is inhibited by GTP, according to ([133] and references cited therein).

The GDH reaction in vitro at neutral pH thermodynamically favors reductive amination of α-ketoglutarate [133]. However, in the brain, formation of α-ketoglutarate and ammonium is favored [134], presumably as a result of the high *K*_m_ for ammonium in the presence of low concentrations of ammonium. This low concentration of ammonium is most likely due to the action of glutamine synthetase, which is of high activity in the brain [135] and will efficiently remove endogenously produced ammonium. Tracer studies with [^13^N]-ammonium, delivered to the portal vein of anesthetized rats, showed very rapid incorporation (within seconds) of label into liver l-glutamate, l-aspartate, and urea [136]. Thus, the GDH reaction, coupled to the AspAT reaction, readily permits portal-vein-derived ammonium and l-aspartate nitrogen to be incorporated into urea. On the other hand, several endogenous l-amino acids utilize an α-ketoglutarate-linked transaminase [Equation (7)] coupled to GDH [Equation (9)] in the liver to generate ammonium (a process referred to as transdeamination; Equation (16)) for urea synthesis [137]. The reverse direction of Equation (9) has been termed transreamination.
l-Amino acid + α-ketoglutarate ⇆ α-keto acid + l-glutamate (7)
l-Glutamate + NAD(P)^+^ + H_2_O ⇆ α-ketoglutarate + ^+^NH_4_ + NAD(P)H (9)
**Net**: l-Amino acid + NAD(P)^+^ + H_2_O ⇆ α-keto acid + ^+^NH_4_ + NAD(P)H (16)

Not only are transdeamination and transreamination reactions important in amino acid metabolism in the liver and other tissues, they also play key roles in the metabolism of cancer cells. Indeed, it has been suggested that the GDH reaction [Equation (9)] may be more important than transamination for the conversion of l-glutamate to α-ketoglutarate in many cancers [133]. The presence of l-glutamate favors binding of NADH to the enzyme, perhaps to curtail l-glutamate oxidation, when cellular redox potentials are elevated. As summarized by Plaitakis et al., GDH expression is upregulated in cancer cell lines and in tissues from patients with different neoplasias, including gliomas and leukemias, as well as in breast, lung and colorectal cancers (see [133] and references cited therein). The authors also pointed out that GDH expression levels correlate with poor outcomes, tumor size and metastatic disease (see [133] and references cited therein).

Studies by Jin et al. showed that GDH is important for the production of α-ketoglutarate from l-glutamate in several human cancer cell lines [138]. Interestingly, stable knockdown of GDH1 in these cancer cell lines, but not in human fetal lung fibroblast MRC-5 and human keratinocyte HaCaT (controls), resulted in decreased cell number [138]. The authors also found that an inhibitor of GDH1 (namely R162; 2-allyl-1-hydroxy-9,10-anthaquinone; 1-hydroxy-2-(2-propen-1-yl)- 9,10-anthracenedione) attenuated cancer cell proliferation and tumor growth in a number of human cancer cell lines but not in control cell lines [138]. R162 was also found to be relatively non-toxic to normal tissue but attenuated tumor growth in an H1299 (lung cancer) xenograft mouse model [138]. The authors concluded that their findings provide proof-of-principle suggesting GDH1 as a promising therapeutic target in the treatment of human cancers associated with elevated l-glutamine metabolism [138]. The same group later showed that GDH1 is upregulated upon detachment of LKB1 (liver kinase B1)-deficient lung cancer cells via pleomorphic adenoma gene 1 (PLAG1), providing anti-anoikic and pro-metastatic properties to these cells [139].

The abovementioned studies highlight the importance of GDH in providing anaplerotic α-ketoglutarate in tumors. However, under hypoxic conditions, as often occurs in the center of solid tumors, the mitochondrial NADH/NAD^+^ ratio increases as a result of slowing of electron transport and concomitant slowing of the rate of NADH oxidation [140]. In addition to inhibiting NADH-producing reactions in the TCA cycle, the high NADH/NAD^+^ ratio will inhibit α-ketoglutarate production in the mitochondria via the GDH reaction. Accordingly, in the presence of l-glutamate, NAD(H) binds more tightly to GDH than NADP(H) does. On the other hand, the overall GTωA reaction [Equation (4)] does not require NAD(P)^+^ and bypasses the GDH reaction as a source of α-ketoglutarate. Thus, *provided there is adequate α-keto acid and l-glutamine present, the GTωA pathway, unlike the GLS1-GDH pathway, can theoretically operate under extremely hypoxic conditions in tumors*.

Critical to tumor progression is the need for cancer cells to overcome oxidative stress induced by high NADH/NAD^+^ ratios. Mitochondrial nicotinamide nucleotide transhydrogenase (NNT) functions as an anti-oxidative enzyme that re-establishes homeostatic balance to regenerate NADPH from NADH. NNT interfaces between the NADH and NADPH pools in mitochondria and contributes 45% of the total NADPH pool by catalyzing the reaction NADH + NADP^+^ ⇆ NADPH + NAD^+^ [141]. NNT is overexpressed in certain cancers and helps protect against oxidative stress associated with glucose deprivation or anchorage-independent growth. Studies of gastric cancer cells show that NNT mediates anoikis resistance, which is believed to protect cells against oxidative stress during anchorage-independent growth [142]. Thus, a shift in NADH to NADPH would support a high NADPH/NADP^+^ ratio and shift the reaction toward increased l-glutamate production from α-ketoglutarate in the mitochondria via the GDH reaction, as shown in Equation (17). (Note that Equation (17) is similar to Equation (9) but is depicted to emphasize the production of l-glutamate with NADPH as reductant). The resulting l-glutamate can be metabolized back to α-ketoglutarate via conversion to l-glutamine followed by entry into the GTωA pathway, circumventing alterations in the NADP^+^ pools.
α-Ketoglutarate + ^+^NH_4_ + NADPH → l-glutamate + NADP^+^ + H_2_O (17)

## 10. “Glutamine Addiction” and the GTωA Pathway in Cancer Cells

### 10.1. Sequence of Enzymatic Reactions in the GTωA Pathway

In contrast to the canonical pathway, in which a deamidation reaction is followed by transamination (or an oxidative deamination reaction), the GTωA pathway involves a transaminase reaction followed by a deamidation reaction. Equations (2)–(4) are rewritten below to emphasize this point. As noted above, important features of the GTωA pathway are that it is irreversible, operates in both cytosolic and mitochondrial compartments, and can theoretically operate under highly hypoxic conditions.
**l-Glutamine** + α-keto acid ⇆ α-ketoglutaramate (KGM) + l-amino acid **Transamination**
(2)
KGM + H_2_O → **α-ketoglutarate** + NH_4_^+^ **Deamidation**
(3)
Net: **l-Glutamine** + α-keto acid + H_2_O → **α-ketoglutarate** + l-amino acid + NH_4_^+^ **GTωA Pathway**
(4)

### 10.2. Why Has the GTwA Pathway Been Largely Overlooked by Biochemists/Oncologists?

In this review, we have presented strong evidence that the GTωA pathway is an important mechanism for producing anaplerotic α-ketoglutarate in normal and cancerous tissues. This leads to the question “why has this pathway been largely ignored since its discovery more than 70 years ago?” Enzyme-catalyzed transamination was discovered by Alexander Braunstein in Russia in the late 1930s. Braunstein and colleagues discovered AspAT and emphasized the metabolic importance of l-glutamate and α-ketoglutarate in enzyme-catalyzed transamination reactions. (A discussion of Braunstein’s early studies on transaminases, which were published in Biokhimiya, can be found in reference [143]). Later work by Esmond Snell (see [144] and references cited therein) and Alton Meister [145] established vitamin B_6_ in the form of PLP as the coenzyme in transamination reactions.

When Meister and colleagues discovered transamination of l-glutamine in the early 1950s [17,18], AspAT was already the most prominent transaminase under study by biochemists, followed by AlaAT. It quickly became apparent after the discovery of AspAT and AlaAT that these enzymes have well-defined roles in amino acid and energy metabolism. Although, as noted above, transaminases such as AspAT do not exhibit absolute substrate specificity, rates of transamination reactions catalyzed by AspAT and AlaAT with l-amino acids/α-keto acids other than their major substrates are generally relatively low compared to those with major substrates. On the other hand, it should be obvious from the above discussions that the l-glutamine transaminases exhibit a very broad l-amino acid/α-keto acid specificity. As a result, at first it was difficult to assign specific biochemical roles to l-glutamine-dependent transaminases, and these enzymes were “overshadowed” by l-glutamate-utilizing transaminases, which exhibit greater substrate specificity. Moreover, both l-glutamate and α-ketoglutarate have long been commercially available, and methods for measuring transamination reactions involving these substrates were established in research and clinical laboratories at least 50 years ago. In contrast, the product of l-glutamine transamination (i.e., KGM) is commercially unavailable and, until very recently, could only be obtained by oxidation of l-glutamine at 37 °C by snake venom l-amino acid oxidase, in the presence of catalase, at neutral pH [19,24]. However, this method presents with some limitations. At neutral pH, l-glutamine slowly cyclizes to 5-oxoproline (5-OP) with the elimination of ammonium, requiring incubations to be rapid and carefully monitored. Nevertheless, earlier preparations of KGM obtained by oxidation of l-glutamine inevitably contained some 5-OP, as well as traces of α-ketoglutarate and l-glutamate (see the discussion in [4]).

Organic procedures for the synthesis of KGM [146,147] and KGM lactam in a vanadium complex [148] are now available—see Ref. [149]. It is anticipated that the availability of pure KGM product will prompt greater attention to the study of l-glutamine transaminases in health and disease and assist in the detailed study of the ω-amidase-catalyzed conversion of KGM to α-ketoglutarate and ammonium [149].

## 11. Glutaminase II (GTωA) Pathway Enzymes in Cancer

### 11.1. Background

Many articles have been published (e.g., [150,151,152,153,154,155,156,157]) that discuss the canonical pathway for l-glutamine conversion to α-ketoglutarate in glutamine addiction (i.e., l-glutamine → l-glutamate ⇆ α-ketoglutarate) in cancer cells. However, due mostly to reasons provided in Section 10.2. above, only perfunctory attention has been accorded the GTωA pathway (i.e., l-glutamine ⇆ KGM → α-ketoglutarate). Several authors (e.g., [158,159,160,161]) have quoted our work that demonstrates GLS1 protein/activity is increased in aggressive prostate cancer cells (cf., [1]), but neglected to provide the rationale associated with the upregulation of the GTωA pathway in these cells. This omission resulted in providing only partial understanding of the integrative relationship between the GTωA and canonical pathways for conversion of l-glutamine to α-ketoglutarate, particularly in cancerous tissues.

Nevertheless, there are a few articles in the cancer literature that specifically mention the enzymes of the GTωA pathway in cancer cells (e.g., [19,162]). Moreover, recently, Thul et al. demonstrated the presence of GTK (annotated as KYAT1/CCBL1) and GTL (annotated as KYAT3/CCBL2) proteins in typical cancer cells such as A-431, U-251MG, and U2OS [163]. The Human Protein Atlas documents that the mRNA for *KYAT1* (i.e., GTK) is present in 17 human cancers, with the highest level in liver cancer (https://www.proteinatlas.org/ENSG00000171097-KYAT1/pathology (accessed on 22 July 2023)). Weak to moderate immunohistochemical staining was performed for 20 human cancers, with strong staining in a few cases of lymphomas and breast and colorectal cancers. Negative staining was observed in a few skin and gastric cancers. The Atlas also documents that mRNA for *KYAT3* (i.e., GTL) is present in all 17 human cancers listed (https://www.proteinatlas.org/ENSG00000137944-KYAT3/pathology (accessed on 22 July 2023)). Immunohistochemical staining of 20 human cancers showed mostly moderate positivity. Thyroid papillary adenocarcinomas, carcinoids, and a few cases of ovarian and endometrial cancers showed strong staining. Thul et al. also demonstrated the presence of Nit2 (i.e., ω-amidase) in typical cancer cell lines, such as A-431, U-251MG, and U2OS [163]. The Human Protein Atlas (https://www.proteinatlas.org/ENSG00000114021-NIT2/pathology (accessed on 22 July 2023)) shows that tumors derived from 16 human tissues all have detectable mRNA for Nit2/ω-amidase, with notably high levels in kidney cancer. The Human Protein Atlas also notes that Nit2 protein can be detected by immunohistochemical staining in a variety of human tumors, with the highest staining intensity in liver and kidney cancers. A recent review article discusses the glutaminase II/GTωA pathway in l-glutamine addiction in cancer [164]. It is hoped that the present article will stimulate more researchers into monitoring the glutaminase II/GTωA pathway as a major contributor to l-glutamine addiction.

Since knowledge of reliance on l-glutamine is a prerequisite for understanding the genetics, biology, and microenvironment in many cancerous as well as in normal cells, it is imperative that investigators in the field provide a comprehensive analysis of the pathways governing l-glutamine metabolism. The requirements for l-glutamine in cancer and normal cells are highly heterogeneous and involve controlled integration between the canonical GLS1 pathway and the GTωA pathway, in association with ancillary pathways that include l-methionine salvage and polyamine formation. Thus, in conclusion, despite being largely ignored in the cancer field, there is now considerable evidence that l-glutamine transaminases and ω-amidase mRNAs and the corresponding proteins are represented in a very wide range of human cancers. We will discuss our findings related to these enzymes and cancer separately below.

### 11.2. Is Nit2/ω-Amidase a Tumor Suppressor or Promoter?

Nit1 (nitrilase-like protein 1) is an enzyme that hydrolyzes the α-keto acid analogue of glutathione (GSH) (designated as deaminated GSH (dGSH)) to α-ketoglutarate and l-cysteinylglycine [165]. The dGSH presumably arises through non-specific transamination of GSH; Nit1 is thus a repair enzyme that “corrects” this “mistake” [165]. Nit1 has been well characterized as a tumor suppressor (e.g., [166,167,168]). Based on moderate sequence homology between Nit1 and Nit2 proteins, Lin et al. suggested that Nit2 might also be a tumor suppressor [52]. Indeed, the authors showed that ectopic expression of Nit2 in HeLa cells inhibited cell growth. This inhibition was through G(2) arrest rather than through apoptosis [52]. However, it has been shown that Nit2/ω-amidase may actually be a tumor promoter in colon cancer [169] and in tongue squamous cell carcinoma [170]. Moreover, folate deficiency (which is associated with increased risk of colon cancer in humans) in human colonocytes in culture resulted in a remarkable ~98% loss of Nit2 (i.e., ω-amidase) protein, as assessed by staining intensity of the Nit2 protein spot on a 2D gel [171]. A possible explanation for this finding is that Nit2 is phosphorylated under low-folate conditions and therefore moves to a different position on the 2D gel. In this context, a change in the isoelectric point of Nit2 compared to that in normal cells was noted in a proteomic study of MCF7 breast cancer cells, consistent with phosphorylation, but only in those cancer cells overexpressing ERBB2 (Erb-B2 receptor tyrosine kinase) [172]. Thus, the role of Nit2/ω-amidase in promoting or suppressing cancer is controversial and needs to be further investigated, as does the possible role of Nit2/ω-amidase phosphorylation in these processes. Nevertheless, despite these uncertainties, we suggest that, on balance, ***ω-amidase benefits tumor growth by providing anaplerotic α-ketoglutarate***.

### 11.3. The GTωA Pathway in Pancreatic Cancer and in Medulloblastoma Tumors

We noted above that KGM is detected in patient-derived pancreatic JHU094 tumors and that the level of KGM increases upon inhibition of GLS1 [71]. The increase in KGM is presumably due at least in part to elevated levels of l-glutamine in the GLS1-inhibited cells, which will favor increased transamination of this amino acid. The disposition of label in metabolites after exposure to ^13^C-labeled l-glutamine suggested the sequence of carbon flow shown in Equation (18) [71]. Isotopologue analyses and GLS1 inhibitor studies showed that this pathway for the formation of l-glutamate from l-glutamine is distinct from that of the simple glutaminase reaction [Equation (1)] [71]. At first glance, Equation (18) (top part) would seem to be a convoluted way to convert l-glutamine to l-glutamate, but the reversible conversion of α-ketoglutarate to l-glutamate offers an explanation for this “convolution”. As noted above, the AspAT reaction is of high activity in most tissues and also presumably in most cancers (e.g., [121]). Thus, l-glutamate plus α-ketoglutarate may be regarded as a single pool of 5-carbon units. α-Ketoglutarate generated by ω-amidase will rapidly equilibrate with l-glutamate in both the cytosolic and mitochondrial compartments. In the mitochondria, the conversion of α-ketoglutarate to succinyl-CoA by the KGDHC in the TCA cycle will ensure directionality to the GTωA pathway, facilitating anaplerosis.
l-Glutamine ⇆ KGM → α-ketoglutarate ⇆ l-glutamate → other 5-C compounds
↓(18)
Succinyl-CoA → TCA cycle

Researchers at Johns Hopkins University have also investigated glutaminolysis in human medulloblastomas [173]. As discussed in [173], medulloblastomas are the most common malignant brain tumors in children, and survival depends on the molecular genetics and epigenetics of the patient’s tumor (see [174,175] and references cited therein). Patients with *Myc*-amplified medulloblastomas exhibit poorer prognoses compared to those patients exhibiting non-*Myc* amplification. In this context, *Myc* promotes the upregulation of GLS1 and thereby promotes cancer cell survival [174,175,176,177,178]. By investigating the isotopomer patterns in l-glutamate generated from l-[U-^13^C]glutamine, Pham et al. deduced that the D425MED orthotopic human *Myc*-amplified medulloblastoma tumor preferentially uses the GTωA pathway over the glutaminase 1 (GLS1) enzyme [Equation (1)] to convert l-glutamine to l-glutamate [Equation (18)] [173]. In support of these findings, the authors noted that *KYAT* (i.e., the gene for GTK) and its mRNA are upregulated in medulloblastoma compared to other pediatric brain tumors in the Children’s Brain Tumor Network/KidsFirst Pediatric Brain Tumor Atlas RNAseq dataset [173].

In conclusion, elegant metabolomic and tracer studies have recently highlighted the importance of the GTωA pathway in selected cancers. It is anticipated that such studies in the future will provide further support for the importance of the GTωA pathway in cancer glutamine addiction.

## 12. Studies from the Authors’ Laboratories Showing the Importance of the GTωA Pathway in Glutamine-Addicted Cancers, with Special Reference to Prostate Cancer

### 12.1. Background

We have reported that normal human pancreatic, bladder, and prostate cells exhibit intense immunohistochemical staining for ω-amidase/Nit2 and GTK/KAT1 [1,32]. We also showed that prostate and bladder cancer cells exhibit intense staining for GTK and ω-amidase, and human pancreatic cancer cells intensely stain for GTK [1,32]. Recently, we investigated the GTωA pathway in normal and cancerous prostate tissue [1]. The specific activity of ω-amidase in normal rat prostate (KGM as substrate; pH 8.5) was found to be about 2/3 that noted for liver and kidney in the same cohort of rats [1]. Interestingly, when we assayed for l-glutamine transaminase activity using three different methods, the specific activity in the rat prostate was found to be considerably higher than that noted for normal rat liver and kidney [1]. (Of the tissues investigated in rats, the highest specific activities of ω-amidase and l-glutamine transaminases were previously noted in the liver and kidney [22,23]). These data strongly suggest that the GTωA pathway is well represented in rat prostate. The presence of GTωA enzymes in human prostate cancer cells is discussed further in the next section.

### 12.2. The GLS1 and GTωA Pathway Enzymes in Human Prostate Cancer Cells in Culture

We investigated the correlation of the intensity of Western blots obtained for GLS1, ω-amidase, and GTK in four human prostate cancer cell lines, with increasing mitotic indices [1]. Among the cell lines investigated, the aggressiveness and propensity to establish bone metastases increases as follows: LNCaP < C4 < C4-2 < C4-2B. This is reflected by increasing intensity of Western blot bands for GLS1, ω-amidase, and GTK (Figure 4). Similar findings were recently reported by Pan et al. [179] and Zhang et al. [180].

An effective mechanism for anaplerosis is especially important for normal and cancerous prostate. Normal human prostate has a very high capacity to synthesize and excrete citrate into the semen [181]. For example, concentrations of citrate in semen and expressed prostatic secretions from non-cancerous human males have been reported to be 132 ± 30 and 222 ± 55 mM, respectively [181]. The concentrations were reported to be significantly lower in semen and expressed prostatic secretions in human males with prostatic cancer (48 ± 8 and 82 ± 36 mM, respectively) [181] but are still quite notable. In normal prostate, aconitase is inhibited by the very large amounts of zinc in this tissue, so that a large portion of citrate generated in the TCA cycle is not directed toward α-ketoglutarate but is, instead, excreted. In contrast, in adenocarcinomatous tissues and in prostate intra-epithelial neoplastic foci, malignant cells exhibit lower concentrations of zinc due to decreased expression of the hZIP1 zinc uptake transporter gene and protein [182]. This results in diminishing the inhibition of aconitase and less citrate secretion, allowing net citrate production to be redirected from the TCA cycle toward synthesis of lipids, cholesterol, steroid hormones, and biomass [183,184]. Malignant prostate cells do not exhibit the specialized function of citrate production and secretion but must adjust metabolic pathways toward energy-efficient citrate-oxidizing cells required for malignancy. Thus, the enormous need for anaplerosis in normal and cancerous prostate is satisfied, at least in part, by an avid uptake system for l-glutamine [185] coupled with the high activities of GLS1 and enzymes of the GTωA pathway [1].

In addition to citrate, semen contains large amounts of polyamines (e.g., [186,187,188,189,190]). Polyamines are known to bind to DNA and help regulate transcription (e.g., [190] and references cited therein). Closure of the methionine salvage pathway by means of the GTωA pathway (Equation (6)) ensures that potentially scarce methyl and sulfur are conserved in the prostate after loss of polyamine carbon to the ejaculate and thus helps to preserve the pool of polyamines necessary for DNA interactions in that organ. We have previously suggested that l-glutamine metabolism can be linked to the methionine salvage pathway via a process that we have labeled the glutamine–methionine bicycle [1,3,4]. In Figure 5a we show how the glutamine–methionine bicycle in normal prostate may regulate the production of citrate, while at the same time conserving sulfur and methyl moieties lost during polyamine biosynthesis. In Figure 5b we show the disposition of citrate and polyamines in relation to the glutamine–methionine bicycle in cancerous prostate.

### 12.3. Expression of GLS1 and GTωA Pathway Enzymes in Relation to Metabolic Cross-Talk between Supporting Stromal Cells and Prostate Cancer Cells

We found high expression levels of GLS1, ω-amidase, and GTK proteins in the glandular architecture of the normal human prostate but negligible expression in the stromal cell compartment. In contrast, we found that as the Gleason score increased from 3 to 5 in biopsied prostate cancer specimens, the GLS1 staining in the glandular compartment was maintained, but staining for GLS1, ω-amidase, and GTK in the stromal cell compartment now became apparent and increased with increasing Gleason score [1]. (The Gleason score (grade) is a measure of the aggressiveness of the tumor cell (https://www.cancerresearchuk.org/about-cancer/prostate-cancer/stages/grades (accessed on 22 July 2023)). The higher the score, the more aggressive is the cancer). We presented evidence that the staining for these enzymes in the stromal compartment is not due to infiltration with cancer epithelial cells but is rather due to the appearance of endogenous enzyme proteins in the stromal cells, whose intensity increases with Gleason score. In conclusion, the canonical pathway (via GLS1) for α-ketoglutarate formation and the alternative and/or compensating GTωA pathway (i.e., l-glutamine transaminase plus ω-amidase) are very highly represented in normal as well as in malignant human prostate.

Although increased gene expression of GLS1, ω-amidase, and GTK proteins in the stromal compartment could not be ruled out, we suggested that the enzymes may have originated from large extracellular vesicles (LEVs) [1]. In this context, Dorai et al. previously showed that metastatic human prostate cancer cells actively discharge LEVs into the tumor microenvironment and that these LEVs contain GLS1 [193]. Whether these LEVs also contain GTK, GTL, and ω-amidase remains to be determined. LEVs have the opportunity to enter the tumor microenvironment and fuse with stromal cells, thereby metabolically reprogramming (or ‘rewiring’) this compartment (cf. [194,195,196,197,198,199,200,201,202]). Considerable evidence suggests that this rewiring of metabolism in stromal cells contributes to cancer malignancies [194,195,196,197,198,199,200,201,202]. Our results suggest that as the Gleason score increases, rewiring of l-glutamine metabolism in the stromal cell compartment contributes to advancing malignancy. We have proposed that the concentration of l-glutamine-derived α-ketoglutarate in the stromal cells is decreased by reactive oxygen species (ROS) emanating from the cancer cells. This loss of anaplerotic α-ketoglutarate, coupled with increased ammonium production in the stromal cell compartment, contributes to enhanced autophagy in these cells. (Ammonium is a potentially toxic and stress-inducing product of amino acid (including l-glutamine) catabolism, which induces autophagy in an mTOR-independent manner to support cancer cell survival [203]). The increased autophagy in the stromal epithelial compartment results in increased release of nutrients to the cancer cell environment, which supports increased cancer cell proliferation.

## 13. l-Glutamine Addiction and Metabolic Plasticity in Cancer Cells

### 13.1. Facets of Metabolic Plasticity—I. Integration of GLS1 and GTωA Pathways

In the following sections, for convenience, we use the term GLS1 pathway to represent the canonical pathway for l-glutamine conversion to α-ketoglutarate (i.e., l-Glutamine → l-glutamate ⇆ α-ketoglutarate). As noted above, earlier work from our laboratory indicated that enzymes of the GTωA pathway are fully represented in rat tissues [1,3,22,23,32]. Immunohistochemistry experiments using human cancer tissues (prostate, bladder, and pancreatic carcinomas) also showed that the malignant tissues express GLS1, GTK, and ω-amidase in significant quantities [1,32]. Previous studies by others have shown that the availability of nutrients is limited inside a growing tumor, particularly within the core microenvironment [204,205,206]. These findings and those from our group have enabled us to hypothesize that the GLS1 and GTωA pathways co-exist in cancer cells and that the relative use of these pathways noticeably contributes to the metabolic plasticity observed in these cells. This concept may be extrapolated to the tumor microenvironment (TME)—see also Section 14 below. This metabolic flexibility may be an important factor in the ability of cancer cells to rapidly replicate. Our studies also predict that these two pathways may be metabolically linked. To appreciate this connection, we review here the metabolism of l-leucine, a well-known activator of mTOR signaling, as a substrate of the BCAT isozymes [Equation (19)]. Leucine influx by the large-neutral amino acid transporter 1 (LAT1) is coupled with ASCT2 (alanine, serine, cysteine-preferring transporter 2), which mediates uptake of l-glutamine. The functional coupling of these transporters facilitates l-glutamine entry into cancer cells. (Note that the α-keto acid substrate of the BCAT isozymes (forward direction) is α-ketoglutarate). The resulting α-ketoisocaproate may be substituted into the general equation for the GTωA pathway [Equation (4)] to yield Equation (20). Thus, l-leucine metabolism may be inexorably linked to controlling the metabolic fate of α-ketoglutarate generated by both pathways. For recent discussions of BCAA metabolism in cancer, refer to [207,208,209].
l-Leucine + α-ketoglutarate ⇆ α-ketoisocaproate + l-glutamate (19)
l-Glutamine + α-ketoisocaproate + H_2_O → α-ketoglutarate + l-leucine + ^+^NH_4_(20)

This interaction between the GLS1 and GTωA pathways may be important in the activation of the mammalian target of rapamycin complex 1 (mTORC1) and in inhibiting autophagy in proliferating cancer cells [210]. In fact, the work of Durán and Hall suggests that mTORC1 senses and is activated by l-glutamine and l-leucine via glutaminolysis and α-ketoglutarate production upstream of *Rag* (recombinant activating gene) [211].

The metabolic plasticity of cancer cells is evidenced by the alternating use of l-glutamine by the two pathways. Metabolic plasticity within the same cancer cell population is well known (e.g., [212,213]). Gradients in l-glutamine concentration are known to occur in cancer cells [213]. It is important to note here that LAT1 mediates the influx of the neutral, essential amino acid l-leucine into cells in exchange for the efflux of intracellular l-glutamine, thus acting as an amino acid antiporter. Variable use of the two pathways provides an explanation for the metabolic gradients that may occur in cancer cells with respect to l-glutamine utilization. Overall, however, these pathways may be optimized in cancer cells toward an enhanced utilization of l-glutamine, providing a new meaning to the “glutamine addiction” of cancers. In other words, the GLS1 pathway may be integrated into the GTωA pathway at least in part by means of l-leucine metabolism.

### 13.2. Facets of Metabolic Plasticity—II. Role of Branched-Chain Amino Acids in Cancer and Their Relationship to the GLS1 and GTωA Pathways

The previous section discusses one BCAA, namely l-leucine. In this section, we discuss the three branched-chain amino acids that occur in proteins. Branched-chain amino acids are essential for cancer growth, as they provide building blocks for protein synthesis. They activate the mTORC1 complex, which, in turn, activates protein translation, growth, and survival. When assimilated, they provide nitrogen for some non-essential amino acids via linked aminotransferases and ultimately for nucleotide biosynthesis. In contrast to l-leucine, which is purely ketogenic, the carbon skeleton of l-valine is purely glucogenic, and that of l-isoleucine is mixed glucogenic/ketogenic. In fact, the first enzymes in the branched-chain amino acids catabolic pathway in the cytosol and mitochondria (BCAT1 and BCAT2; Section 9.2.3) are overexpressed in several cancers. This overexpression may serve as a diagnostic indicator and a prognostic marker for certain cancers [207,208,209]. Since BCAAs are essential, cancers rely either on dietary intake or scavenging them from apoptotic/necrotic cells for survival and progression. We suggest that there is a metabolic interplay not only between the GLS1 pathway and BCAA metabolism in cancer cells but most notably between the GTωA pathway and BCAA metabolism that contributes to the metabolic plasticity of cancer cells.

The relationship between BCAA metabolism and the GLS1/GTωA pathways is depicted diagrammatically in Figure 6, which highlights the important role that BCAAs play in regulating α-ketoglutarate metabolism in cancer (cf., [214]). As an amino group acceptor, α-ketoglutarate levels are reduced by high BCAT levels in different types of cancer cells. Raffel et al. showed that increased branched-chain amino acid degradation by BCAT1 overexpression is required for proliferation, survival, and maintenance of cancer cell stemness in acute myeloid leukemia both in vitro and in vivo via restriction of α-ketoglutarate levels [215]. Therefore, in acute myeloid leukemia cells (and possibly other cancer cells), the interaction of BCAA metabolism with that of the GLS1 and GTωA pathways results in fine-tuning of the levels of α-ketoglutarate needed for cancer progression via increased BCAA metabolism.

### 13.3. Facets of Metabolic Plasticity—III. Potential Role of ω-Amidase and l-Asparagine in Metastatic Colonization

Distant metastatic lesions are under constant metabolic stress, and, therefore, the cancer cells in their new residences must adapt to an otherwise hostile microenvironment. In order to survive, they undergo intense metabolic reprogramming [216,217]. The GTωA pathway described herein provides a mechanism for the metastatic cancer cell to rewire its metabolism so that it can survive and grow in a hostile environment. An important event in the metastatic cascade is the initiation process of epithelial-to-mesenchymal transition (EMT), a developmental process by which the proliferating cancer cells acquire mesenchymal properties that facilitate their migration, invasion, and establishment of micro-metastases. During this transition that ensures survival, cap-dependent protein synthesis is significantly reduced, and cells are directed to synthesize stress-responsive proteins through alternative cap-independent (Internal Ribosome Entry Site (IRES)-mediated) translation mechanisms. Accordingly, Knott et al. have demonstrated that, in a mouse model of breast cancer, the non-essential amino acid l-asparagine governs the formation of metastatic lesions through regulation of the EMT [218]. This is accomplished primarily by an upregulation of asparagine synthetase (ASNS) [Equation (21)]. l-Asparagine and l-glutamine metabolism are intricately linked because l-asparagine amide nitrogen is derived from l-glutamine amide nitrogen [Equation (21)]. Notice that the ASNS reaction can act as an alternative to the GLS reaction for the formation of one equivalent of l-glutamate from one equivalent of l-glutamine, but since two high-energy bonds of ATP are lost, the ASNS reaction is much more energetically expensive than is the GLS1 reaction.
l-Aspartate + l-glutamine + ATP → l-asparagine + l-glutamate + AMP + PP_i_(21)

Proteomic studies have revealed that an elevated asparagine content occurs in proteins that drive the EMT **[219]**. Moreover, limiting l-asparagine bioavailability compromised the expression of upregulated proteins in EMT at both the translational and transcriptional levels. Furthermore, limiting l-asparagine (and also l-glutamine) bioavailability by exogenous administration of an asparaginase/glutaminase inhibitor downregulates β-catenin activity, resulting in decreased stemness of cells that are necessary for “seeding” metastatic lesions [219]. (Overexpression of β-catenin in the nucleus is known to stimulate the production of various oncogenes [220]). It is also known that as extracellular l-glutamine levels fall, tumor cells adapt and become dependent on l-asparagine for proliferation and protein synthesis [221]. The role of l-asparagine in the adaptation of the cancer cell to l-glutamine deprivation is further evidenced by (1) its capacity to upregulate the expression of glutamine synthetase (GLUL), (2) its capacity to suppress apoptosis under l-glutamine deprivation, and (3) its ability to activate mTORC1 activity [222,223].

Normal concentrations of l-asparagine in the cellular environment are relatively low compared to those of l-glutamine. For example, concentrations of l-asparagine and l-glutamine in human plasma are <0.1 mM and ≥0.5 mM, respectively (e.g., [224]). This relatively low concentration of l-asparagine suggests that this amino acid could be a sensitive regulator of cancer proliferation and the EMT. Indeed, l-asparagine depletion by a knockdown of ASNS or treatment with bacterially expressed asparaginase reduces breast cancer invasion and metastasis without affecting the growth of the tumor at the primary site [218]. But how are l-asparagine levels normally regulated in cancer cells? One mechanism for altering l-asparagine levels is to lower the concentration of l-glutamine (as a source of l-asparagine amide nitrogen) by inhibiting glutamine synthetase.

However, cells may be capable of depleting l-asparagine through the rarely considered asparaginase II pathway. This pathway is analogous to the glutaminase II/GTωA pathway, except that l-glutamine is replaced by l-asparagine [Equations (22)–(24)] [225]. Transamination of l-asparagine with a suitable α-keto acid generates KSM, which, as noted above, is also a substrate of ω-amidase. The oxaloacetate thus formed can feed into the TCA cycle. Stimulation of asparagine removal through the asparaginase II pathway may be clinically useful. For example, provision of a suitable α-keto acid for transamination of l-asparagine such as glyoxylate will drive the reaction toward l-asparagine utilization. Glyoxylate is an especially good substrate for transamination of l-asparagine [226]. Moreover, transamination of glyoxylate is almost irreversible. However, because glyoxylate is also a substrate of the glutamine transaminases, a balance of the anti-cancer effect of lowering asparagine concentration against the pro-cancer effect of increased activity of the GTωA pathway must be considered. This is an area of research that requires further study.
l-Asparagine + α-keto acid ⇆ α-ketosuccinamate (KSM) + l-amino acid(22)
KSM + H_2_O → oxaloacetate + ^+^NH_4_(23)
Net: l-Asparagine + α-keto acid + H_2_O → oxaloacetate + l-amino acid + ^+^NH_4_(24)

### 13.4. Facets of Metabolic Plasticity—IV. Role of GLS1/GTωA Pathways in Tumor–Stroma Collaboration

Oncogenic signaling pathways and cancer epigenetics alone cannot sustain cancer progression [227]. Cancer progression, however, is sustained by signals and paracrine loops, involving, for example, gradients of epidermal growth factor and colony-stimulating factor that activate tumor stromal cells (cancer-associated fibroblasts; CAFs) in the vicinity (e.g., [228]). Once activated, these CAFs provide cancer cells with nutrients such as amino acids, fatty acids, carbohydrates, lactic acid, and ketone bodies by increasing autophagy within the CAF compartment, thereby contributing to the increased biomass in the cancer cell compartment. This “host-parasite relationship” has given rise to the concept of “two-compartment metabolism” that involves catabolic CAFs and anabolic cancer cells, which is characteristic of the metabolic heterogeneity among tumors [229].

The authors of recent studies have extended the “two-compartment” concept by proposing that cancer cells “instruct” adjacent stromal cells to alter their metabolic pathways to support the mitotic activity of tumor cells. CAFs contribute to the “parasitic” activity of cancer cells by upregulating glutamine synthetase to compensate for l-glutamine deprivation within the tumor core [230,231]. Thus, cancer cells can “highjack” CAFs and reprogram their metabolic machinery to increase l-glutamine production within the microenvironment of the tumor where l-glutamine levels are depleted. l-Glutamine is assimilated by tumor cells and metabolized by the GLS1 pathway to provide anaplerotic α-ketoglutarate [229,230]. The enhanced production of l-lactate by cancer cells apparently contributes to the metabolic reprogramming of CAFs [230,231]. Increased ammonium production through enhanced GLS1 activity within cancer cells diffuses into CAFs, which results in ammonium-induced autophagy [232]. The availability of l-glutamine fuels this vicious cycle of autophagy, as shown by Ko and coworkers [231]. The “commandeering” of CAFs by tumor cells to adjust their metabolic function is viewed as a “cell state” rather than a novel “cell type” within the tumor microenvironment. Any deviation from their tumor-induced cell state would potentially be of therapeutic significance. Thus, reprogramming CAFs through therapeutic or genomic modification to normal fibroblasts may be an important focus of targeted approaches to control of cancer.

In this context, we have shown that large normal glandular cells stain strongly for ω-amidase in several tissues, including those in the prostate [3,32]. Although the elevated expression of GLS1 in the stromal cell compartments has been recognized for thyroid and breast cancers [233,234], the elevation in GTK and ω-amidase in the stromal cell compartments has, thus far, only been identified within prostatic cancer [1]. Further studies are required to determine whether cancer cell-induced elevation in GTK and ω-amidase extends to stromal cell compartments of other cancerous tissues. The metabolic “flexibility” of the GTωA pathway to supply essential nutrients to cells with minimal energy requirements would markedly contribute to tumor cell aggressiveness and would underscore the importance of these enzymes in catering to the altered metabolic patterns observed in cancer biology.

We hypothesize that the increased expression of GTK, ω-amidase, and GLS1 in activated fibroblasts in the stromal compartment is critical to the metabolic reprogramming that occurs in cancer cells, including prostate cancer cells. We further suggest that these enzymes facilitate the proliferative capacity of cancer cells, promote stromal cell adaptation to the cancer environment, and facilitate self-destructive autophagy, which contributes to a phenomenon known as the “Reverse Warburg Effect”. The reverse Warburg effect was proposed in 2009 by Pavlides et al., who suggested that during tumorigenesis, the hypoxic condition in the stromal cells results in the production of l-lactate and pyruvate that are taken up by aerobic cancer cells as an energy source [235]. However, the exact molecular mechanisms whereby GLS1 and GTωA participate in the collaborative nutrient exchange between tumor and stroma remain to be elucidated.

## 14. Will Inhibitors of GTωA Enzymes, Perhaps in Combination with Inhibitors of GLS1, Glutamine Transporters, or Other Proteins/Enzymes, Be Clinically Useful in Cancer Patients?

A comprehensive list of drugs targeting l-glutamine metabolism has recently been compiled by Vanhove et al. [236]. Chemically reactive small-molecule analogues of l-glutamine, such as 6-diazo-5-oxo-l-norleucine (DON), azaserine, and azotomycin (NSC-56654), although limited in their capacity, were first considered as potential anti-cancer agents more than 65 years ago ([237,238,239] and references cited therein). These nucleophilic drugs generally act as l-glutamine mimetics, covalently targeting the catalytic l-serine or l-cysteine moiety in the active site of l-glutamine amidohydrolases such as GLS1 or of glutamine amidotransferases [240,241]. Although earlier animal studies and phase II clinical trials with nucleophilic l-glutamine mimetic agents (e.g., [242,243,244]) appeared promising, there has been no recent continuation of clinical studies with these agents due to unacceptable side effects (see [241] for a discussion of DON clinical trials). Healthy, rapidly dividing intestinal cells rely heavily on l-glutamine metabolism for energy production and for building biomass [245]. Thus, a marked limitation in DON’s therapeutic efficacy is that it targets all rapidly dividing cells, which include mucosal-epithelia-producing gastrointestinal problems. In order to overcome this problem, Lemberg et al. at Johns Hopkins School of Medicine designed two prodrugs of DON that can be administered in doses that are much lower than those of unmodified DON but are effective and can bypass the intestine [241]. The prodrug strategy for DON was shown to be effective in an animal model of *Myc*-driven human medulloblastoma [246,247].

In Section 3, we noted that the l-glutamine transaminases exhibit broad specificity toward l-amino acids and α-keto acids. Thus, since DON [N_2_CHC(O)CH_2_CH_2_CH(CO_2_^−^)(NH_3_^+^)] is a mimic of l-glutamine [NH_2_C(O)CH_2_CH_2_CH(CO_2_^−^)(NH_3_^+^)], it has the potential to bind effectively as an l-amino acid substrate to the l-glutamine transaminases. Indeed, it has been found that DON is an l-amino acid substrate of human GTK [149]. Interestingly, azaserine is a β-lyase substrate of human GTK [149]. Because ω-amidase (an amidohydrolase) contains a crucial l-cysteine residue in the active site [21,56,248], it is predicted that the α-keto acid formed by transamination of DON will be a powerful, irreversible, mechanism-based inhibitor of ω-amidase. Whether (a) transamination of DON contributes to the anti-cancer effects of this compound by generating the corresponding α-keto acid and whether (b) this α-keto analogue can be synthesized in vitro as a potential ω-amidase *k*_cat_ inhibitor remain to be established [149].

We noted above that BPTES is an allosteric inhibitor of GLS1 (and GAC). The Johns Hopkins group have encapsulated BPTES in nanoparticles and shown that, in combination with metformin, it is effective against pancreatic cancer in a mouse model [249]. The Cerione group at Cornell University has been instrumental in studying the mechanism of inhibition of GLS1/GAC by BPTES and in characterizing additional compounds related to BPTES, which may prove useful in future clinical trials [73,74,75,76]. According to Milano et al., over 2000 BPTES analogues have been synthesized [76]. Two of these BPTES analogues that exhibit strong inhibition of GLS1/GAC (i.e., CB-839 [250] and IPN60090 [251]) are undergoing Phase I clinical trials as anti-cancer agents. Calithera Biosciences, a clinical-stage oncology biopharmaceutical company that specializes in targeted therapies for biomarker-specific patient populations, has developed CB-839 and erlotinib, which together reduce glucose and glutamine uptake, for treating patients with EGFR (epidermal growth factor receptor)-mutated stage IV non-small-cell lung cancer [252].

Jin et al. [253] have recently summarized mechanisms involved in the resistance of cancer cells to glutaminase inhibitors. Such mechanisms include (a) increased uptake of and reliance on glucose and fatty acids as an energy source, (b) increased uptake of aspartate, which in turn, leads to increased TCA cycle metabolite malate, (c) internalization of macromolecules that can supply amino acids, and (d) increased expression of p53, which in turn leads to expression of proteins that help maintain energy and redox balance. These mechanisms could presumably be targeted by suitable inhibitors. In addition, it has been suggested that *N*-aspartylglutamate may be an energy “reservoir” that can be used in place of glutamine [254]. In that case, an inhibitor of carboxypeptidase II may be a useful anti-cancer agent in combination with a GLS1 inhibitor.

Udupa et al. have shown that genetic suppression of GTK in pancreatic tumors (P198 shGTK-KD cancer cells injected in the back of nude mice) leads to *complete* suppression of pancreatic tumorigenesis [71]. The authors suggested that current trials testing the efficacy of GLS1 inhibitors as anti-cancer agents should take into account the presence of pathways other than that initiated by GLS1 for metabolism of l-glutamine. The authors suggested that a GTK inhibitor may be useful, either alone or in combination with a GLS1 inhibitor, for the treatment of cancer. But are there inhibitors of GTK that have already been developed? Transition state (TS) mimetics that inhibit KMB transamination in cells in culture have been developed, including l-methionine ethyl ester pyridoxal (MEEP) [255,256]. KMB is a substrate of both GTL and GTK. Therefore, it is probable that these TS mimetic inhibitors will block the l-glutamine transaminases. Interestingly, MEEP was shown to induce DNA strand breaks in HeLa cells, which is typical of apoptotic cell death [256]. Some inhibitors of GTK/KAT1 (based on kynurenine as amino acid transaminase substrate) have been designed with *K*_i_ values ranging from ~20 µM to ~1 mM [51] but, to our knowledge, have not been tested as anti-cancer agents.

Another strategy might be to design a specific inhibitor of ω-amidase that can be used alone or in combination with other anti-cancer drugs, such as an inhibitor of GLS1 or an inhibitor of glutamine transporters. In this context, Wang et al. have demonstrated that the l-glutamine transporter ASCT2 (SLC1A5) is highly expressed in prostate cancer patient samples [257]. The authors used LNCaP and PC3 prostate cancer cell lines to show that chemical or shRNA-mediated inhibition of ASCT2 function in vitro decreases l-glutamine uptake; cell cycle progression through E2F transcription factors; mTORC1 pathway activation, and cell growth [257]. It has been suggested that inhibitors of ASCT2 might be useful anti-cancer agents [258], but, to our knowledge, they have not yet been used in clinical studies. The l-glutamine transporter LAT1 has been shown to be elevated in a number of cancers ([259] and references cited therein). Indeed, a phase 1 clinical trial of a LAT1 inhibitor (Nanvuranlat; JPH203) against biliary tract cancer is currently underway [259]. Another family member (LAT2) is highly expressed in osteosarcoma cells and plays a critical role in tumor immune evasion [260]. It was suggested that an inhibitor of this transporter might be clinically useful [260]. In a recent review, Zhao et al. summarized the changes in seven transporter transcripts (compared to those of normal tissues) (SLC1A5, SLC3A2, SLC7A5, SLC7A8, SLC6A14, SLC38A1, SLC38A2) in 20 human cancers [261]. In the conclusion section, the authors stated that “SLC7A5, SLC7A8, SLC38A1, and SLC38A2 may regulate the polarization of tumor-associated macrophages (TAMs). SLC1A5, SLC3A2, SLC7A5, and SLC6A14 may be promising biomarkers for BC [breast cancer] diagnosis and may represent potential therapeutic targets for these patients” [261].

Major impediments to successful immunotherapy for cancer are the tumor-generated immune escape mechanisms that diminish an effective anti-tumor T-cell response. Cancer cells exert significant control over the TME and thus can exhaust T-cell surveillance by limiting nutrient availability, releasing immunosuppressive cytokines, or creating a hypoxic environment. As noted, cancer cells and immune cells exhibit high demands for glutamine, arginine, and BCAAs to sustain growth and proper functioning [262,263]. Thus, inhibiting transport or utilization of these essential nutrients would be detrimental to metabolism of both cancer cells and cytotoxic T-cells. This conundrum, which limits the availability of essential amino acids in tumor cells, while making them accessible to cytotoxic T-cells, may be resolved by considering targeted regulation of the GTωA pathway. Previous studies have shown that cancer cells capture and “usurp” these essential nutrients at the expense of neighboring stroma and immune cells. We hypothesize that the differential usage of essential nutrients is accomplished in part by a “dual regulation of GTωA” between the cancer cells and the immune cells. Accordingly, relative expression levels and the activity of amino acid transporters, glutaminase, arginase, oxidizing enzymes, mTOR, and BCAA metabolizing enzymes along with GTωA will form a metabolic network of interconnected pathways that can act as potential immune–metabolic control points. Perhaps regulation of the metabolic economy of “supply and demand” may provide an effective strategy for diminishing cancer cell proliferation and enhancing immune function. Such a hypothesis suggests that inhibition of the GTωA pathway, which is highly expressed in cancer cells, by specific inhibitors will “disproportionately” prevent cancer cells from thwarting the immune response and concurrently enable immune cells to achieve immune surveillance.

However, a tight-binding, specific inhibitor of ω-amidase has not yet been developed. We believe that development of such an inhibitor is a priority. A specific ω-amidase inhibitor will inhibit the formation of α-ketoglutarate from KGM (and oxaloacetate from KSM) and thus strongly interfere with the cancer cell’s anaplerotic energy requirement. Such an inhibitor may have advantages over an inhibitor of l-glutamine transaminases. As discussed above, l-glutamine transaminases may be involved in several metabolic processes (e.g., formation of the neuromodulator kynurenine; closure of the methionine salvage pathway; salvage of α-keto acids), whereas ω-amidase has a narrower metabolic role. Moreover, evidence suggests that excess KGM resulting from inhibition of ω-amidase may be relatively benign, as discussed in [149].

## 15. Conclusions

In this review, we have pointed out that the GTωA pathway has been overlooked by biochemists and biomedical scientists. However, there is now overwhelming evidence that this pathway for l-glutamine conversion to α-ketoglutarate (i.e., l-glutamine ⇆ KGM → α-ketoglutarate) is at least as important as the canonical pathway (i.e., l-glutamine → l-glutamate ⇆ α-ketoglutarate) in mammals, including humans. We have suggested that the pathway may be important for (1) salvage of α-keto acids, arising through non-specific transamination reactions, and (2) possible transfer of α-keto acids/l-amino acids between cellular and/or subcellular compartments. We have also provided evidence that the pathway occurs in tumors and proposed that the pathway may be beneficial to tumors because it can provide anaplerotic α-ketoglutarate (and possibly oxaloacetate) even under highly hypoxic conditions. By generating α-ketoglutarate, the GTωA pathway may be especially advantageous in prostate cancers, which require enhanced anaplerosis to meet their energy and biosynthetic demands. The normal and cancerous prostate also generate a considerable amount of polyamines, which require a salvage mechanism to ensure that scarce methyl and sulfur derived from l-methionine are not lost during the polyamine synthesis. Closure of the methionine salvage pathway by the GTωA pathway ensures that l-methionine is generated from KMB, while at the same time providing anaplerotic α-ketoglutarate, even under hypoxic conditions.

Finally, we have suggested that inhibitors of l-glutamine transaminases and especially ω-amidase, either alone or in combination with a GLS1 inhibitor or an l-glutamine transporter may be clinically effective anti-cancer agents. We suggest that enhancement of the immune response in the TME together with an inhibitor of ω-amidase may also be clinically useful. It is hoped that the present review will spur interest in exploring the metabolic roles of the GTωA pathway in normal and cancerous tissues.

## Figures and Tables

**Figure 1 biology-12-01131-f001:**
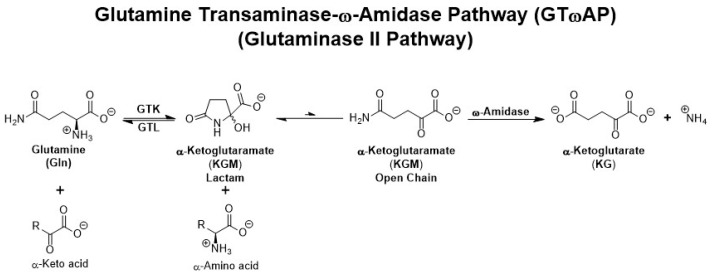
The GTωA (glutaminase II) pathway. Note the involvement of two enzymes—an l-glutamine transaminase and ω-amidase—that are involved in the production of ammonium from l-glutamine. Note also the equilibrium between open-chain (ω-amidase substrate) and lactam forms of KGM. KGM greatly favors the lactam form. GTK and GTL are glutamine transaminases (see Section 3 below).

**Figure 2 biology-12-01131-f002:**
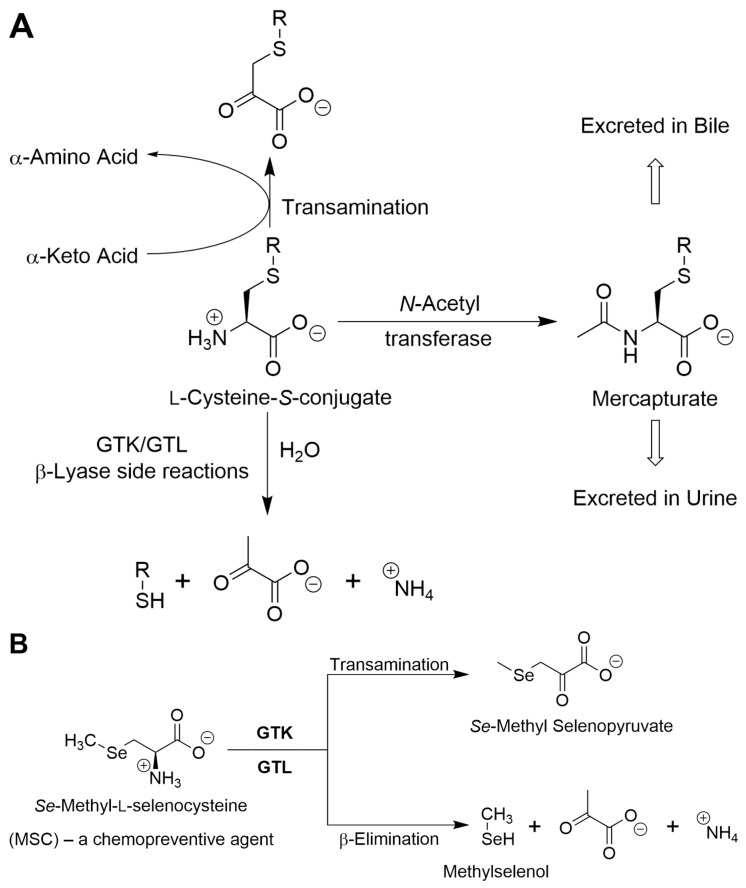
(**A**) Possible fates of l-cysteine *S*-conjugates. These compounds may be transaminated by GTK or GTL to the corresponding α-keto acids or *N*-acetylated by *N*-acetyltransferases to the corresponding *N*-acetyl *S*-conjugates (mercapturates) and excreted. However, if –SR is a good leaving group, the l-cysteine *S*-conjugate may undergo a β-lyase reaction to form pyruvate, ammonium, and a sulfur-containing fragment (RSH). If the RSH fragment is stable, it may be *S*-methylated to RSCH_3_ and excreted—the thiomethyl shunt (not shown). In some cases, the l-cysteine *S*-conjugates (e.g., those derived from halogenated alkenes) are toxic as a result of the extreme reactivity of the eliminated RSH. (**B**) Competing GTK/GTl-catalyzed transamination/β-lyase reactions with the chemopreventive agent *Se*-methyl-l-selenocysteine. For further details see [2,3,33,41,42,45].

**Figure 3 biology-12-01131-f003:**
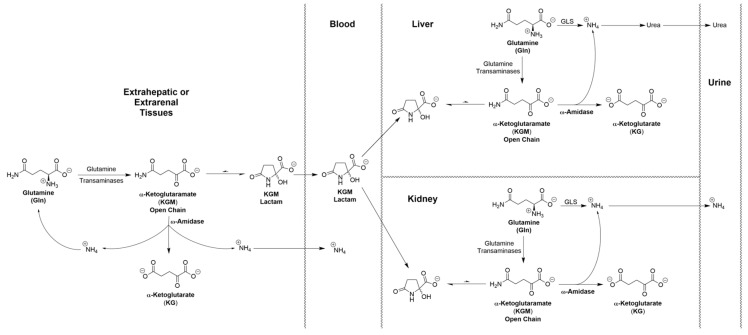
Proposed role of KGM as an “ammonium carrier” from extrahepatic and extrarenal tissues to liver and kidney. Once KGM is taken up by the liver, it enters a large pool of KGM generated by endogenous transamination of l-glutamine. The ammonium generated from this pool of KGM can then be incorporated into urea, into l-glutamine (not shown), or into both. It is suggested that KGM can also enter the kidney. This KGM together with endogenously generated KGM can then act as a source of ammonium, contributing to the role of the kidneys in maintaining acid–base balance. GLS, glutaminase.

**Figure 4 biology-12-01131-f004:**
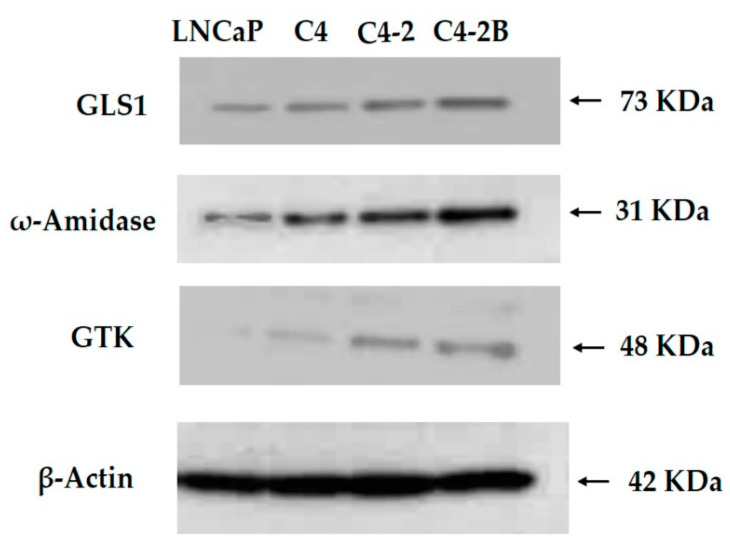
Western blots of homogenates (~25 μg protein in each lane) of cultured human prostate cancer cells. Note that the intensity of the bands for GLS1, ω-amidase, and GTK increases with an increase in their mitotic capacity. The trend was verified by quantitative densitometric analysis. Reproduced from [1] with permission.

**Figure 5 biology-12-01131-f005:**
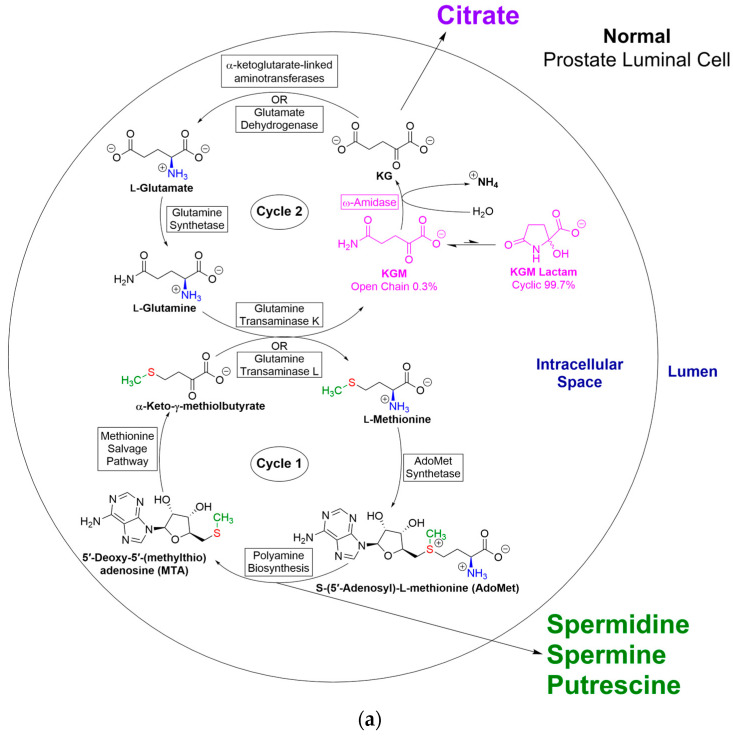
(**a**) Disposition of citrate and polyamines in relation to the glutamine–methionine bicycle in normal prostate. The importance of the GTωA pathway (i.e., l-glutamine transaminases linked to ω-amidase) in closing the methionine salvage pathway is shown in cycle 1. This salvage pathway is especially important in the prostate due to the excretion of high levels of polyamines in the seminal fluid. Cycle 2 shows the importance of the GTωA pathway for the production of anaplerotic α-ketoglutarate, which feeds into the TCA cycle. Under normal conditions, a large amount of the TCA cycle carbon in the prostate is not completely oxidized to CO_2_ but rather is involved in the production of the extraordinarily high concentrations of citrate that are excreted into the seminal fluid. Reproduced from [3] with permission. (**b**). Disposition of citrate and polyamines in relation to the glutamine–methionine bicycle in cancerous prostate. In cancerous prostate, a smaller (but still substantial) amount of citrate is present in the seminal fluid compared to normal prostate. In the cancerous tissue, a relatively large amount of citrate carbon is channeled toward lipid (and cholesterol/steroid hormone) biosynthesis for the cancer cells’ “selfish” biosynthetic needs. In particular, androgen-independent prostate cancer cells can synthesize testosterone directly from cholesterol by upregulation of a biosynthetic pathway ordinarily present in testes and adrenal glands. In addition, inactive adrenal-derived androgens, such as dehydroepiandrosterone and 4-androstene-3,17-dione, can be converted to active androgens (testosterone and dihydrotestosterone) by aldo-keto reductase family 1, member C3 (AKR1C3), typically localized to stromal and prostatic endothelial cells [191,192]. Reproduced from [3] with permission.

**Figure 6 biology-12-01131-f006:**
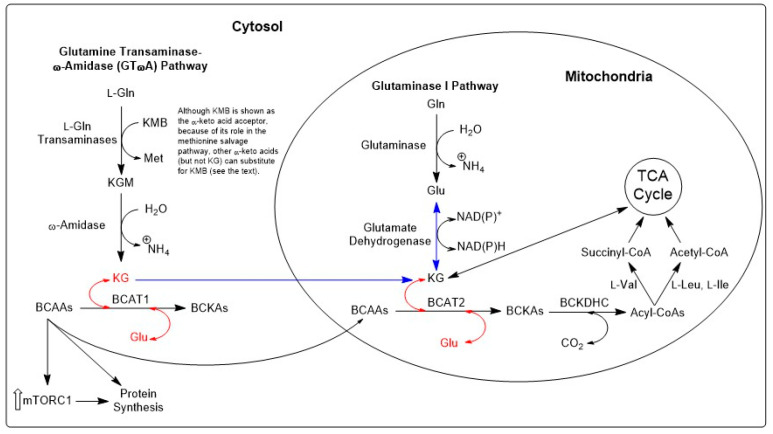
Proposed role of glutaminase I (GLS1) and GTωA pathways in the metabolism of branched-chain amino acids in cancer. The branched-chain amino acids in the cytosol stimulate the mTORC1 and, thus, increase cellular protein synthesis. The BCAAs are also transaminated into the corresponding α-keto acids in the cytosol and mitochondria by the action of BCAT1 and BCAT2, respectively. This process involves the conversion of α-ketoglutarate to l-glutamate (red arrows). The α-keto acids thus generated are converted to acyl-CoAs, which can be further metabolized to succinyl-CoA and enter the TCA cycle to generate useful energy. Thus, there is a ***net consumption of α-ketoglutarate*** by the action of the two BCATs. α-Ketoglutarate can potentially be replenished by the action of ***both*** GLS1 (followed by the GDH reaction (shown) or an l-glutamate-linked aminotransferase (not shown)) and the GTωA pathways (blue arrows). A consequence of this pathway interaction is that GLS1 inhibitors that are currently in human clinical trials such as CB-839 may lack full efficacy because the GTωA pathway can substitute for the GLS1 pathway to generate α-ketoglutarate, which will continue to be transaminated by the branched-chain amino acid aminotransferases. BCAA, branched-chain amino acid; BCAT1, branched-chain aminotransferase 1 (cytosolic); BCAT2, branched-chain aminotransferase 2 (mitochondrial); BCKA, branched-chain α-keto acid; Glu, l-glutamate; GLS1, glutaminase isozyme-1; Gln, l-glutamine; KGM, α-ketoglutaramate; KG, α-ketoglutarate; KMB, α-keto-γ-methylthiobutyrate; mTORC1, mammalian target of rapamycin complex 1; Met, l-methionine; TCA, tricarboxylic acid.

## Data Availability

All reasonable requests concerning this article will be honored.

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
