# Peer review of "Metabolic Heterogeneity, Plasticity, and Adaptation to “Glutamine Addiction” in Cancer Cells: The Role of Glutaminase and the GTωA [Glutamine Transaminase—ω-Amidase (Glutaminase II)] Pathway"

_biology, 2023, doi:10.3390/biology12081131_

Round 1

Reviewer 1 Report

Long, deep article, of great interest and written with great knowledge and experience on the part of the authors, which focuses on the addiction of glutamine in tumor cells and the description, in particular, of the Omega amidase pathway .

-        This reviewer considers the article to be of great interest, written with great knowledge and mastery of the subject, and yet perhaps there is a series of detailed information that exceeds the scope of a scientific article and may correspond much more to the scope of a scientific textbook in hands of the interested researcher and not so much for the clinician or oncologist who is looking for information in relation to the issue of glutamine addiction in cancer.

-        The comment tries to be positive under the view that an extensive and detailed article on biochemical and clinical aspects may have less acceptance and be used by a smaller number of researchers than when the information is provided through its natural channels depending on the subject or the specialty.

Ex: - the two paragraphs contained between lines 821-840 could be summarized in a few lines.

-        - section 11.1 of background (845-906) could perhaps be condensed to make the article a little lighter

-       -  etc…

-        On the other hand, I would like to suggest that authors review all those critical aspects with scientific articles published by other authors on the subject. This reviewer is of the opinion that it is more useful for the reader to stick to the description of the information and try to avoid personal debates and/or criticism of other authors as much as possible.

-        Perhaps it would be good to give a greater degree of coherence to the message: while in line 1033 and following it is stated that another author has misinterpreted the immunohistochemical findings, then in another place it is stated that stromal staining appears with increasing gleason grade (1128 and following).

-        1140 et seq., regarding the origin of enzymes in stromal cells or extracellular vesicles ( LEVs ), seems implausible in this context and the information should be justified or revised.

As minor comments:

-        The suggestion to review the adequacy, length and representativeness of the content of the title of the article .

-        It might be useful to include a list with all abbreviations and enzyme nomenclature included in the article.

-        The role of mitochondria in relation to the described metabolic changes could be considered in a specific way.

-        A section on the role of the microbiota in relation to glutamine metabolism could be of interest

-        Figure three is difficult to observe and could be graphically adapted to make it more comfortable, for example, to the same size as figure 4.

Finally, congratulate the authors for their extensive contribution and in particular for this excellent work.

Author Response

       Long, deep article, of great interest and written with great knowledge and experience on the part of the authors, which focuses on the addiction of glutamine in tumor cells and the description, in particular, of the Omega amidase pathway .

Reply: We thank the reviewer for the encouraging words

 -        This reviewer considers the article to be of great interest, written with great knowledge and mastery of the subject, and yet perhaps there is a series of detailed information that exceeds the scope of a scientific article and may correspond much more to the scope of a scientific textbook in hands of the interested researcher and not so much for the clinician or oncologist who is looking for information in relation to the issue of glutamine addiction in cancer.

Reply: We interpret this statement to mean that such a large review will be of interest to only a small readership of known experts rather than to a wider readership – the larger general audience may have a difficult time understanding the message in the review. We appreciate this point, but it is important that experts in the field of cancer addiction be made aware of the highly integrative nature of the GLS1 and GTωA pathways. Thus, in the revised manuscript we have emphasized even more than previously the importance of the GTωA pathway, especially in the sections containing the Introduction and Conclusion. In addition, in view of the comprehensive nature of the article we have deemed it essential to provide an all-inclusive, unified approach to explain pertinent information that integrates the subject matter pertaining to the GTωA pathway rather than having it disseminated among several articles or journals. Finally, as noted above, KGM is becoming noticed by physicians other than oncologists-        

The comment tries to be positive under the view that an extensive and detailed article on biochemical and clinical aspects may have less acceptance and be used by a smaller number of researchers than when        the information is provided through its natural channels depending on the subject or the specialty.

Reply: We interpret this statement to mean that such a large review will be of interest to only a small readership of known experts rather than to a wider readership – the larger general audience may have a difficult time understanding the message in the review. We appreciate this point, but it is important that experts in the field of cancer addiction be made aware of the GTωA pathway.   Thus, in revised manuscript we have emphasized even more than previously the importance of the GTωA pathway, especially in the Introduction and Conclusion.  In addition, in view of the comprehensive nature of the article we have deemed it essential to provide an all-inclusive, unified approach to explain pertinent information that integrates the subject matter pertaining to the GTωA pathway rather than having it disseminated among several articles or journals. Finally, as noted above, KGM is becoming noticed by physicians other than oncologists

Ex: - the two paragraphs contained between lines 821-840 could be summarized in a few lines.

Reply: We agree – this section has been greatly shortened in the revised manuscript.

-        - section 11.1 of background (845-906) could perhaps be condensed to make the article a little lighter

-       -  etc…

Reply: This section has been shortened as requested.

-        On the other hand, I would like to suggest that authors review all those critical aspects with scientific articles published by other authors on the subject. This reviewer is of the opinion that it is more useful for the reader to stick to the description of the information and try to avoid personal debates and/or criticism of other authors as much as possible.

Reply: There are very few articles on the GtωA pathway and we believe that we have covered most of them in the review.  We were surprised by three recent articles that referenced our work but selectively chose portions of our data that paralleled their understanding of the GLS1 pathway and avoided the main thrust of the work that integrates the GLS1 and the GTωA pathway. We were especially troubled by a statement in one article that unfairly criticized one aspect of our work based on a false premise.  Nevertheless, in the revised manuscript we have avoided directed criticism of the three articles in question and systematically addressed any misconception and misinterpretation of data regarding glutamine metabolism. 

Reviewer 2 Report

To Authors:

This reviewer comments/suggestion attached as a PDF copy.

As per the attached comments and suggestions.

Author Response

Summary of the Research:

Authors in these manuscripts summarize the metabolic function of GTωA pathway, which provides some anticancer strategies. In herein, authors specially highlight the glutaminase II pathway related research, including biological roles of the GTωA pathway related research, biological roles including of the GTωA pathway, evidence of GTωA pathway operate extensively in vivo, canonical pathway for α-ketoglutarate formation in cancer et al.  It’s great review for researchers to quickly understand GTωA pathway and related cancer research, however, if authors could further add below contents I mentioned in comments, which will make this review better.

Reply: We thank the referee for stating that our review is a “great review”. We have answered the criticisms listed below.

Major Comments:

1) In the role of glutamine metabolism for cancer treatment, authors could summarize the mechanism

involved in each step of glutamine metabolism, from glutamine transporters to redox homeostasis.

Reply: As we have highlighted in this review, the scientific literature was developing misconceived ideas that the major route for metabolism of glutamine metabolism is via the glutaminase (GLS) pathway (i.e., Gln g Glu g α-ketoglutarate gTCA cycle).  However, as we systematically address in our review the glutaminase II (GTωA) pathway (i.e., Gln g α-ketoglutaramate (KGM) g α-ketoglutarate g TCA cycle) is also very important. We mention other pathways for utilization of glutamine amide nitrogen such as in the synthesis of asparagine and purines/pyrimidines, but quantitatively these pathways are probably minor compared to channeling of glutamine through the glutaminase and GTωA pathways.  This point is made clearer in the revised manuscript. We did mention the important role of glutamine transporters. In the revised manuscript we have placed greater emphasis on the transporters.  Finally, we incorporated supportive  references regarding redox homeostasis.  It has been known for well over a hundred years that α-keto acids are quantitatively decarboxylated by H2O2 [RC(O)CO2- + H2O2 g RCO2- + H2O + CO2]. In this context α-keto acids such as α-ketoglutarate can function as antioxidants by directly converting H2O2 to H2O while at the same time generating CO2 and a metabolizable fatty acid (Liu et al. Biomed. Res. Int. 2018, 3408437, 2018).  In addition some work suggests that α-ketoglutarate can actually stimulate the production of anti-oxidant defenses (He et al. J. Agric. Food Chem. 66, 43, 11273-11283, 2018).   We now include these references along with references to the pathway that incorporates α-ketoglutarate and hypoxia-inducible factor (HIF) 1α which helps maintain metabolic stability in cancer cells.  But the review is already very long and the role of α-keto acids as anti-oxidants is tangential to the main discussion so we have kept this addition short.

2) Authors could also briefly summarize recent clinical data on glutamine metabolism research in

hyperammonemic disease and discuss the role of KGM as a biomarker for hyperammonemic disease.

Reply: In the original review we did discuss KGM as a biomarker for hyperammonemic diseases. We even mentioned the possibility that KGM concentrations may be elevated in cancers that are associated with hyperammonemia.  In the revised manuscript we introduce new ideas concerning KGM and nitrogen/ammonia homeostasis.  It is well known that excess ammonia is neurotoxic.  There are many theories regarding the mechanism of this neurotoxicity and we have now included a reference to Roger Butterfield (an expert on hyperammonemic diseases) who lists some of these theories.  But there is no doubt that hyperammonemia leads to excess formation of glutamine in the astrocyte compartment which is detrimental. Brusilow and colleagues have suggested that excess glutamine in the brain leads to osmotic stress – the osmotic gliopathy theory (Brusilow et al. Neurotherapeutics 7(4) 452-470, 2010). This reference is now included in the revised manuscript.  But again, in order not to lose focus of the main thrust of the review we have kept additions to this section to a minimum.

2) Authors could also discuss the mechanisms underlying cancer cell resistance to agents that target

glutamine metabolism, as well as strategies for overcoming these mechanisms.

Reply: We have just become aware of a very recent and useful review article that summarizes mechanisms involved in the resistance of cancer cells to glutaminase inhibitors (Jin et al. Exp. Mol. Med. 55, 706-715, 2023). Such mechanisms include a) increased uptake and reliance of glucose and fatty acids as an energy source, b) increased uptake of aspartate, which in turn, leads to increased TCA cycle metabolite malate, 3) internalization of macromolecules that can supply amino acids, and d) increased expression of p53, which in turn leads to expression of proteins that help maintain energy and redox balance.  These mechanisms could presumably be targeted by suitable inhibitors.  In addition, Anne Le’s group have suggested that N-aspartylglutamate may be an energy “reservoir” that can be used in place of glutamine (Shen et al. Curr. Opin. Chem. Biol. 62, 64-81, 2021).  In that case an inhibitor of carboxypeptidase II may be a useful anti-cancer agent in combination with a glutaminase inhibitor. These points and appropriate references are now included in the revised manuscript.

3) Authors could add some clinical trial data about combination treatment using the selective inhibitors targeting glutaminase and transaminase, discuss the importance of combined therapy for avoiding metabolomic response.

Reply: In the original manuscript we did suggest that a combination of a GLS1 inhibitor and a glutamine transaminase inhibitor may be clinically useful. However, to our knowledge no inhibitors of glutamine transaminase has been used in clinical trials. We emphasize the “stressing need” for the development of such inhibitors and especially for selective inhibition of ω-amidase.

4) Targeting glutamine transporters as a treatment for cancer, upregulation of glutamine transporters

SLC1A5, SLC38A1, SLC38A2, and SLC6A14 at the cell membrane is required, so authors could summarize cancer treatment strategies focused on pharmacological inhibition of these transporters.

Reply: In the original manuscript we did briefly mention glutamine uptake inhibitors.  As suggested by the reviewer, in the revised manuscript we have expanded our discussion of the upregulation of glutamine transporters and possible strategies for their pharmacological inhibition.  In the revised manuscript we now include a reference to Zhao et al (Ann. Trans. Med 10(14) 777, 2022).  In Table 2 of their manuscript these authors list the changes in seven transporter transcripts (compared to normal tissues) (SC1A5, SC3A2, SC7A5, SC7A8, SC6A14, SC38A1, SC38A2) in 20 human cancers. In the conclusion section the authors stated “SLC7A5, SLC7A8, SLC38A1, and SLC38A2 may regulate the polarization of tumor-associated macrophages (TAMs). SLC1A5, SLC3A2, SLC7A5, and SLC6A14 may be promising biomarkers for the BC diagnosis and may represent potential therapeutic targets for these patients”.

5) Given that activation and differentiation of immune cells are coupled to metabolic reprogramming, regulating the metabolic activity of immune cells should be considered in the development of potential strategies that target glutamine metabolism. Authors could summarize evidence glutamine metabolism in immune cells and how inhibitors of glutamine metabolism elicit antitumor and blockade in the TME.

Reply: In the original manuscript we did mention the role of the TME in supporting the adjacent cancer cells, but did not specifically mention immune cells. In checking PubMed we have come to realize that there is an enormous literature on immune cells in the TME and cancer, especially in regard to breast cancer.  It is beyond the scope of the present review to discuss the role of immune cells in cancer in detail.  Nevertheless, in the revised manuscript we have included a recent review of the subject that nicely summarizes the enhanced reliance of tumors on glutamine, arginine and branched-chain amino acids.  These amino acids are derived in part from immune cells in the TME, thereby depriving these cells of important nutrients leading to immune suppression and reduced cytotoxic killing of the cancer cells (Wetzel et al. Front. Oncol. 13, 1186539, 2023). Thus, inhibitors of glutamine/arginine/branched-chain amino acid uptake/metabolism will not only cause an energy stress in cancer cells but will de-repress the immune cells in the TME. Such an inhibitor in conjunction with an ω-amidase inhibitor may be clinically useful.  

Minor Comments:

To better understand the pathway and networks in glutamine metabolism, authors could draw some

figures.

Reply: In the revised manuscript we have added two more figures.